# Open-channel structure of a pentameric ligand-gated ion channel reveals a mechanism of leaflet-specific phospholipid modulation

John T. Petroff II[1], Noah M. Dietzen[1], Ezry Santiago-McRae [2], Brett Deng[1], Maya S. Washington[1], Lawrence J. Chen[1], K. Trent Moreland [1], Zengqin Deng[3,4], Michael Rau [5], James A. J. Fitzpatrick [3,5,6,7], Peng Yuan [3,4], Thomas T. Joseph [8], Jérôme Hénin [9], Grace Brannigan [2,10] & Wayland W. L. Cheng [1]✉

Pentameric ligand-gated ion channels (pLGICs) mediate synaptic transmission and are sensitive to their lipid environment. The mechanism of phospholipid modulation of any pLGIC is not well understood. We demonstrate that the model pLGIC, ELIC (*Erwinia* ligand-gated ion channel), is positively modulated by the anionic phospholipid, phosphatidylglycerol, from the outer leaflet of the membrane. To explore the mechanism of phosphatidylglycerol modulation, we determine a structure of ELIC in an open-channel conformation. The structure shows a bound phospholipid in an outer leaflet site, and structural changes in the phospholipid binding site unique to the open-channel. In combination with streamlined alchemical free energy perturbation calculations and functional measurements in asymmetric liposomes, the data support a mechanism by which an anionic phospholipid stabilizes the activated, open-channel state of a pLGIC by specific, state-dependent binding to this site.

pLGICs such as the nicotinic acetylcholine receptor (nAChRs) are ubiquitous in the nervous system where they determine synaptic transmission and neuronal excitability[1]. These channels are targets of general anesthetics and anti-epileptics[2], and potential targets for the treatment of many neuropsychiatric diseases[3–5]. pLGIC function is modulated by certain lipids[6], including anionic phospholipids which enhance agonist response in the nAChR[7–11]. Anionic phospholipids, as well as sterols and fatty acids, have also been shown to modulate other pLGICs[12–16]. It is intuitive that lipids modulate pLGICs by influencing the conformational stability of different states that the channel samples during gating; however, mechanisms of phospholipid modulation of any pLGIC remain poorly understood.

[1]Department of Anesthesiology, Washington University School of Medicine, Saint Louis, MO, USA. [2]Center for Computational and Integrative Biology, Rutgers University, Camden, NJ, USA. [3]Department of Cell Biology and Physiology, Washington University School of Medicine, Saint Louis, MO, USA. [4]Center for the Investigation of Membrane Excitability Diseases, Washington University School of Medicine, Saint Louis, MO, USA. [5]Center for Cellular Imaging, Washington University School of Medicine, Saint Louis, MO, USA. [6]Department of Neuroscience, Washington University School of Medicine, Saint Louis, MO, USA. [7]Department of Biomedical Engineering, Washington University School of Medicine, Saint Louis, MO, USA. [8]Department of Anesthesiology and Critical Care, Perelman School of Medicine, University of Pennsylvania, Philadelphia, PA, USA. [9]Laboratoire de Biochimie Théorique, Institut de Biologie Physico-Chimique, Université Paris Cité, CNRS UPR 9080, Paris, France. [10]Department of Physics, Rutgers University, Camden, NJ, USA. ✉e-mail: wayland.cheng@wustl.edu

With the recent abundance of lipid-bound pLGIC structures by cryo-EM, investigation into the structural mechanisms of lipid modulation has focused on direct binding of lipids to specific sites[12,13,15,17–22]. Structures of the 5-HT$_{3A}$ receptor (5-HT$_{3A}$R), GABA$_A$ receptor, nAchR and the prototypic pLGICs, *Erwinia chrysanthemi* ligand-gated ion channel (ELIC) and *Gloeobacter violaceus* ligand-gated ion channel (GLIC), show bound phospholipids at various sites[14,19,23–28]. However, a phospholipid density in a structure is insufficient evidence for a modulatory mechanism[6]. Establishing such a mechanism demands knowledge of the functional effect of the phospholipid, identity of phospholipids that occupy the site, and state-dependent binding at that site consistent with the modulatory effect of the phospholipid. Recent structural studies of ELIC show a phospholipid and cardiolipin density bound to different sites in the apo or resting structure, but functional analysis indicates that cardiolipin increases the open probability of the channel[14,26]. Another study of the 5-HT$_{3A}$R shows evidence of state-dependent phospholipid binding, but the identity of this phospholipid in the structure and the modulatory effect of phospholipids in 5-HT$_{3A}$R is unknown[23].

In this study, we determine a structural mechanism for anionic phospholipid positive modulation of the pLGIC, ELIC. An open-channel structure of ELIC shows a phospholipid density in an outer leaflet intersubunit site not well resolved in non-conducting structures. The phospholipid binding site shows a distinct conformation in the open-channel structure. Along with streamlined free energy perturbation calculations[29] and functional measurements in asymmetric proteoliposomes the results support a mechanism by which the anionic phospholipid, phosphatidylglycerol (PG), stabilizes the activated, open-channel state of ELIC by binding to a specific site.

## Results

### Modulation of ELIC by neutral and anionic phospholipids

pLGICs such as the nAchR require an anionic phospholipid and phosphatidylethanolamine (PE) for maximal agonist response[7]. Using a fluorescence stopped-flow liposomal flux assay, we previously showed that ELIC agonist responses are significantly enhanced in 2:1:1 POPC (1-palmitoyl-2-oleoyl-phosphatidylcholine): POPE (1-palmitoyl-2-oleoyl-phosphatidylethanolamine): POPG (1-palmitoyl-2-oleoyl-phosphatidylglycerol) liposomes compared to POPC-only liposomes[13,30]. This effect manifested as an increase in peak response, and decrease in the rate and extent of desensitization (Fig. 1a, b). However, the exact contributions of POPG and POPE on these effects were not clear.

To delineate the respective effects of these lipids, ELIC activity was measured in binary mixtures of 3:1 or 1:1 of POPC:POPE or POPC:POPG. Responses were measured using 10 mM propylamine (a saturating concentration of agonist, Supplementary Fig. 1a), since cysteamine (the agonist used previously[13]) at concentrations >5 mM is incompatible with the permeating ion, thallium (Tl$^+$). Only POPG (25–50 mol% POPG) slowed desensitization (Fig. 1c, d). Accordingly, POPG increased relative channel activity at 10 and 25 s after agonist application (Supplementary Fig. 2). This finding is consistent with patch-clamp experiments showing that POPG decreases the extent of desensitization in ELIC[13,26]. Interestingly, both POPG and POPE were required to achieve maximal responses (Fig. 1d), indicating that both lipids are needed to facilitate channel activation. However, the EC$_{50}$ for the peak agonist dose-response in POPC-only liposomes was not significantly different than in 2:1:1 POPC:POPE:POPG liposomes (Supplementary Fig. 1). Taken together, the results indicate that POPG modulates ELIC by stabilizing the activated, open-channel state relative to desensitized states[13,26]. In the presence of POPE, POPG also enhances channel peak response, which may result from an increase in gating efficacy or an effect on desensitization. While differences in channel reconstitution and orientation in the various liposome compositions could also contribute to the effect on peak response, experiments using asymmetric liposomes described below control for these factors, verifying that POPG increases peak channel activity. Furthermore, no significant difference was appreciated in the size of POPC and 2:1:1 POPC:POPE:POPG liposomes used for the assay (Supplementary Fig. 3).

### Capturing an open-channel conformation of ELIC

To investigate the mechanism of positive modulation by POPG, we determined structures of WT ELIC by single particle cryo-electron microscopy (cryo-EM) in the absence and presence of agonist in MSP1E3D1 nanodiscs with 2:1:1 POPC:POPE:POPG (hereafter referred to as 2:1:1) or POPC (Supplementary Fig. 4, Supplementary Table 1). For all structures, 10 mM cysteamine was used as the agonist since this is a saturating concentration of the most potent, known agonist of ELIC (Supplementary Fig. 6b)[31]. The POPC and 2:1:1 WT ELIC apo structures were indistinguishable. Likewise, the agonist-bound structures were identical in both lipid conditions (Supplementary Fig. 5), deviating insignificantly from apo and propylamine-bound structures in POPC nanodiscs previously reported[14,32]. Importantly, the agonist-bound structures showed a non-conducting pore with occlusion at 16' (F247) and 9' (L240) (Fig. 2e, f). Thus, structures of WT ELIC in poorly activating POPC compared to activating 2:1:1 lipids exhibited no meaningful structural differences.

Since POPG stabilizes the activated, open-channel state of ELIC, we aimed to capture an open-channel conformation using gain-of-function mutations. We began with the previously reported gain-of-function triple mutant V261Y/G319F/I320F (ELIC3)[33] that introduces three aromatic residues at the top of the M4 helix (M4) (Supplementary Fig. 6a). As previously reported, ELIC3 caused a left-shift in the agonist dose response curve, slowed desensitization in the presence of agonist, and slowed deactivation as assessed by excised patch-clamp recordings from 2:1:1 giant liposomes (Supplementary Fig. 6b–e). Interestingly, the structure of agonist-bound ELIC3 in 2:1:1 nanodiscs, determined at 3.3 Å resolution (Supplementary Fig. 4, Supplementary Table 1), showed opening of the pore at 16' but still occlusion at 9' (Fig. 2d–f). Seeking a mutant that remains open in the presence of agonist, we added the mutations P254G (previously shown to slow deactivation[34]) and C300S (previously shown to slow desensitization in the presence of agonist[12,35]), and produced P254G/C300S/V261Y/G319F/I320F (ELIC5) (Supplementary Fig. 6a). Strikingly, ELIC5 showed a deactivation rate ~1000× slower than WT and no evidence of desensitization in the presence of 10 mM cysteamine (Fig. 2a, Supplementary Fig. 6e), indicative of profound stabilization of the activated, open-channel state. In four recordings of ELIC5 currents in response to 10 mM cysteamine where single channel openings were resolved, there was no evidence of channel closure for several minutes (Supplementary Fig. 6g), suggesting that ELIC5 has a high open probability at steady state. Consistent with its gain-of-function phenotype, a structure of ELIC5 in 2:1:1 nanodiscs with 10 mM cysteamine, determined at 3.4 Å resolution (Supplementary Figs. 4 and 7, Supplementary Table 1), yielded an apparent open pore (Fig. 2d–f) with a pronounced outer leaflet phospholipid-shaped density (Fig. 2c).

The structures of WT, ELIC3 and ELIC5 reveal different pore conformations that indicate a possible sequence of changes from a resting to open-channel conformation (Fig. 2d, e). The WT apo structure shows constrictions at 16' and 9' (Fig. 2e), consistent with resting pore conformations reported for the nAchR[19,28,36]. In the presence of agonist, the top of M2 in WT shows a modest blooming of 16'. The ELIC3 gain-of-function mutant further opens at 16' with the F247 side chain turning away from the pore into an intersubunit space, but otherwise resembles the WT structure. Finally, in ELIC5, 9' rotates away from the pore into an intersubunit space and 16' opens further, such that the narrowest diameter of the pore is at 2' (Q233) (Fig. 2f). Two rotameric states of Q233 could be modeled in the structure yielding a minimum pore diameter of 5.6 Å (rotamer 1, ELIC5 CA 1) and 7.3 Å (rotamer 2, ELIC5 CA 2) (Fig. 2f and Supplementary Fig. 8). The larger pore diameter is consistent with the capacity of ELIC to conduct large

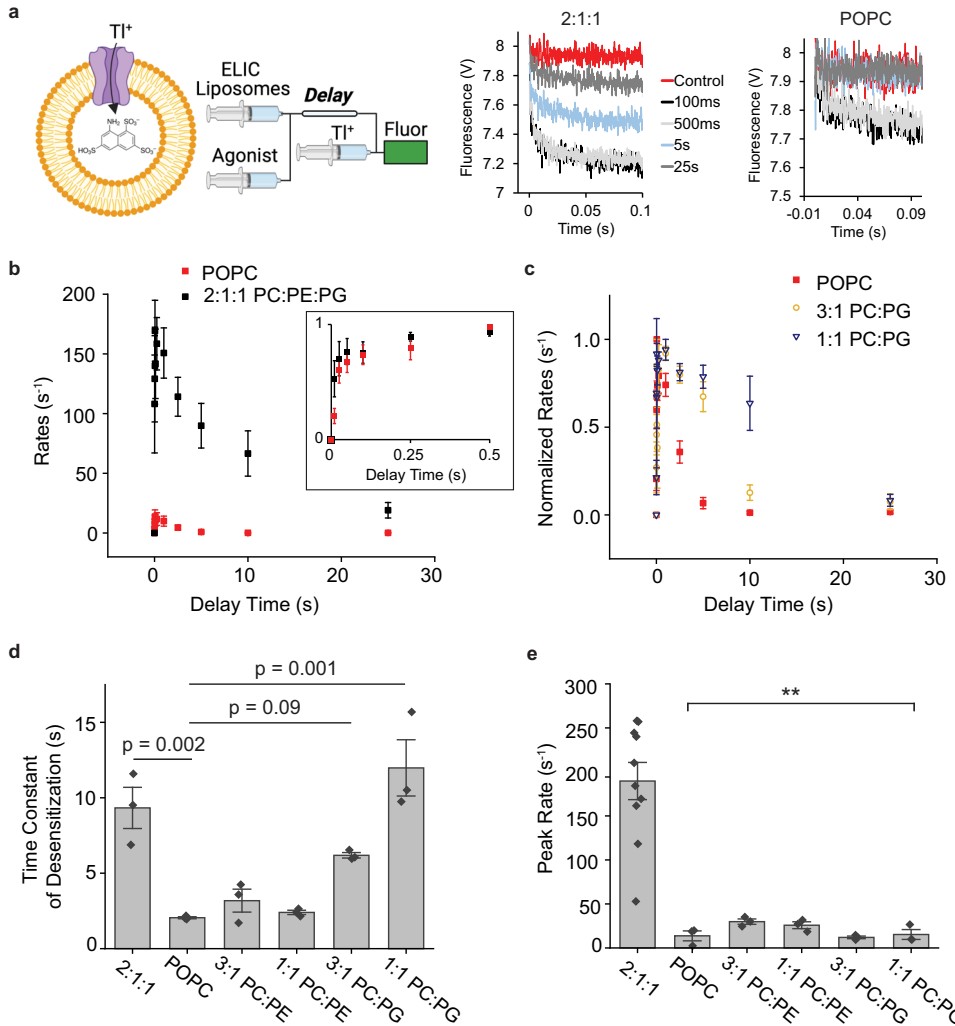

**Fig. 1 | ELIC agonist responses in different lipid conditions. a** Left: Schematic of the sequential mixing experiment for the fluorescence liposomal stopped-flow assay (created with BioRender.com). ELIC proteoliposomes containing the fluorescent Tl[+] indicator, ANTS, are mixed with agonist (propylamine), and then with Tl[+] solution after a variable delay time. *Right*: Representative raw fluorescence traces; the first 0.1 s are fit with a stretched exponential to determine the fluorescence quenching rate, which relates to ELIC channel activity. The legend shows control (no agonist) and different delay times. **b** Tl[+] flux rates of WT ELIC in 2:1:1 POPC:POPE:POPG (black) and POPC (red) liposomes as a function of time after mixing with 10 mM propylamine ($n = 3$). Inset shows activation time course of normalized data. Activation was too fast to accurately determine the rate. **c** Normalized Tl[+] flux rates of WT ELIC in POPC, 3:1 POPC:POPG and 1:1 POPC:POPG as a function of time after mixing with 10 mM propylamine ($n = 3$). **d** Weighted time constant for the time course of desensitization in response to 10 mM propylamine from the Tl[+] flux assay for the indicated liposome compositions ($n = 3$). 2:1:1 indicates 2:1:1 POPC:POPE:-POPG. **e** Same as **c** showing peak channel activity (rate) in response to 10 mM propylamine from the Tl[+] flux assay (2:1:1, $n = 12$; other, $n = 3$). Statistical analysis was performed using a one-way ANOVA and post-hoc Tukey test. ** indicates $p < 0.01$ when comparing 2:1:1 with all other liposome compositions. *P*-values for these comparisons with 2:1:1 are 0.003 for POPC, 0.006 for 3:1 PC:PE, 0.005 for 1:1 PC:PE, 0.002 for 3:1 PC:PG and 0.003 for 1:1 PC:PG. Data are shown as mean ± se for (n) independent experiments. Source data are provided as a source data file.

quaternary cations[37]. ELIC5 has the same single channel conductance as WT (Fig. 2b, Supplementary Fig. 6f); this along with its high open probability strongly suggests that the ELIC5 structure represents a functionally-relevant open-channel conformation. The fact that the pore diameters at 16′ (12.6 Å) and 9′ (13 Å) in the ELIC5 CA structure are much larger than at 2′ is consistent with previous findings that mutations of 9′ and 16′ do not affect ion conduction[37]. Further work will be necessary to assess the contribution of Q233 rotameric states to ion conductance and selectivity[38–40].

The patch-clamp and Tl[+] flux data of WT and ELIC3 show robust activation and desensitization in 2:1:1 membranes such that agonist-bound structures of WT and ELIC3 are expected to show desensitized conformations. However, the observed structures are most consistent with pre-active conformations based on structures of other pLGICs, especially the nAchR, where 16′ and 9′ appear to be activation gates and desensitization consistently involves a constriction in the most

intracellular aspect of the pore[28,32,35,36]. It is possible that some unknown factor influencing the cryo-EM structures is producing this apparent discrepancy or that these structures actually represent desensitized conformations in ELIC. Therefore, we cannot confidently assign the WT and ELIC3 agonist-bound structures as pre-active or desensitized conformations, and further work is necessary to clarify their annotation. Hereafter, to examine POPG modulation of ELIC, we will focus on WT apo, WT agonist-bound (WT CA) and the ELIC5 agonist-bound (ELIC5 CA) structures all in 2:1:1 nanodiscs, which are putative resting, agonist-bound non-conducting and open-channel conformations, respectively.

## ELIC channel opening is associated with an outer leaflet bound phospholipid
The ELIC5 CA structure shows a strong phospholipid density at an outer leaflet site within an intersubunit groove formed by M4, M3 and

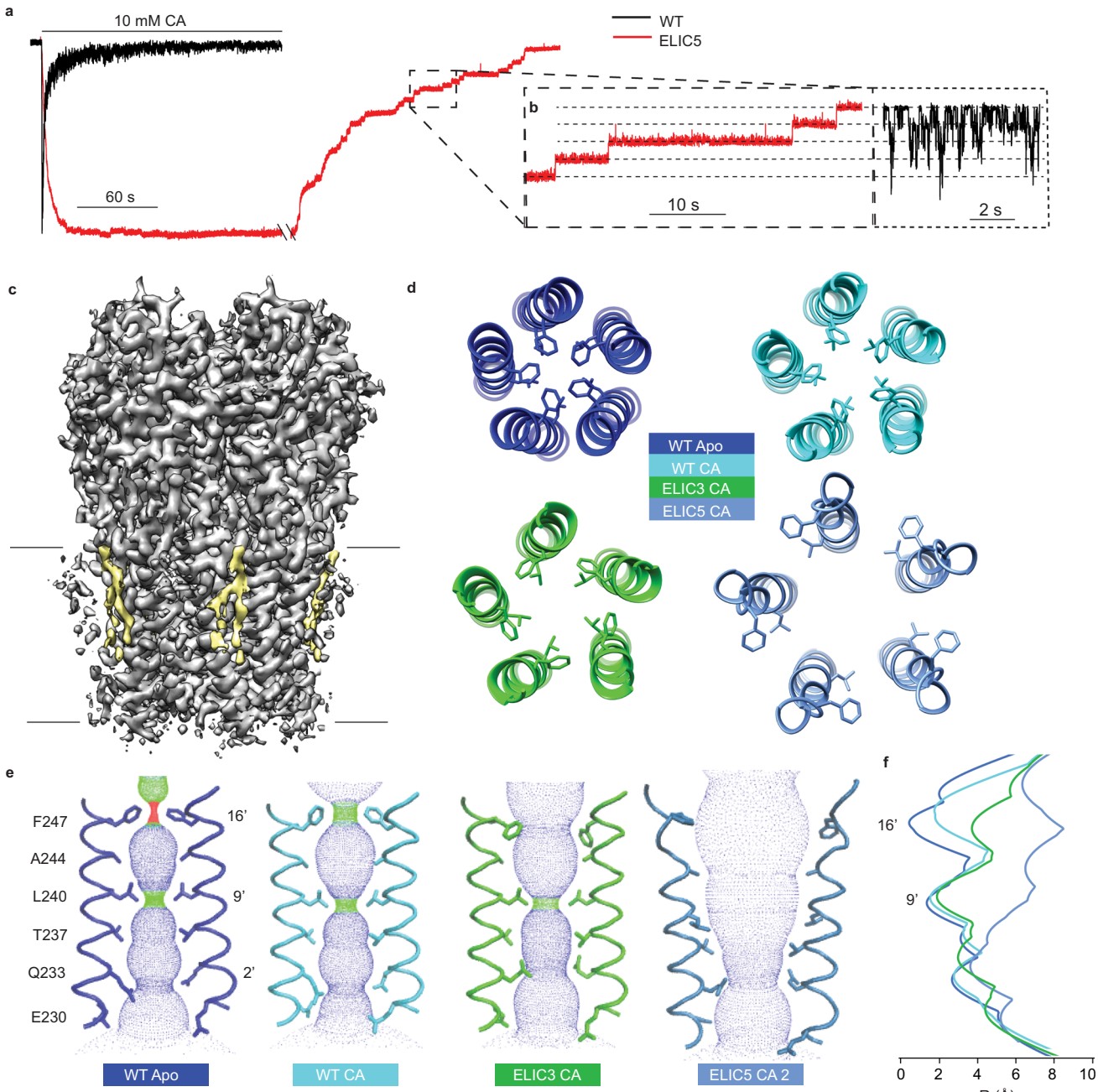

**Fig. 2 | Conformational changes of pore opening. a** Representative ELIC currents in response to 10 mM cysteamine of excised patches from 2:1:1 POPC:POPE:POPG giant liposomes (black = WT, red = ELIC5, −60 mV holding voltage). The ELIC5 currents show sustained response to 10 mM cysteamine, followed by slow deactivation with the removal of cysteamine. **b** Inset that shows ELIC5 single channel currents taken from the deactivation trace (i.e., channel closure) or WT (black) single channel currents in the presence of 10 mM cysteamine. **c** Cryo-EM density map of ELIC5 in 2:1:1 POPC:POPE:POPG nanodiscs + 10 mM cysteamine (grey) with outer leaflet lipid density (yellow) and black lines approximating the bilayer. **d** View of the pore lining M2 helix from the extracellular side showing 16′ (F247) and 9′ (L240) side chains from WT apo, WT CA (WT + 10 mM cysteamine), ELIC3 CA (ELIC3 + 10 mM cysteamine), and ELIC5 CA (ELIC5 + 10 mM cysteamine). All are structures in 2:1:1 POPC:POPE:POPG nanodiscs. **e** Ion permeation pathway as determined by HOLE[76] of the same structures shown in **d**. The ELIC5 CA structure shows rotamer 2 for Q233 (see Supplementary Fig. 8 for a comparison of rotamers 1 and 2). Labeled are 16′, 9′, and 2′ side chains which form the narrowest portions of the pore in different structures. **f** Pore radius as a function of distance along the pore axis for the same structures as **d** and **e**.

M1 (Fig. 3a). The density has well-resolved features including nearly full density for the palmitoyl-oleoyl acyl chains. Of the six structures obtained in this study, three others show evidence of phospholipid density at this outer leaflet site, including WT apo structures in POPC and 2:1:1, and the WT agonist-bound structure in 2:1:1 (Supplementary Fig. 9a)[14]. Thus, among agonist-bound structures, phospholipid density at this site was not observed in POPC nanodiscs. To compare the strength of the phospholipid densities in 2:1:1 structures, we low pass

filtered the maps to 3.5 Å resolution and displayed the map for each lipid at a sigma value of 3.0 (Fig. 3b). The maps show relatively weak lipid density in the WT CA structure compared to the ELIC5 CA structure, suggesting higher occupancy of a phospholipid in the open-channel conformation relative to the agonist-bound non-conducting conformation at this site.

The phospholipid binds at an outer leaflet site with the headgroup located below the β6-β7 loop and adjacent to the M2-M3 linker, and

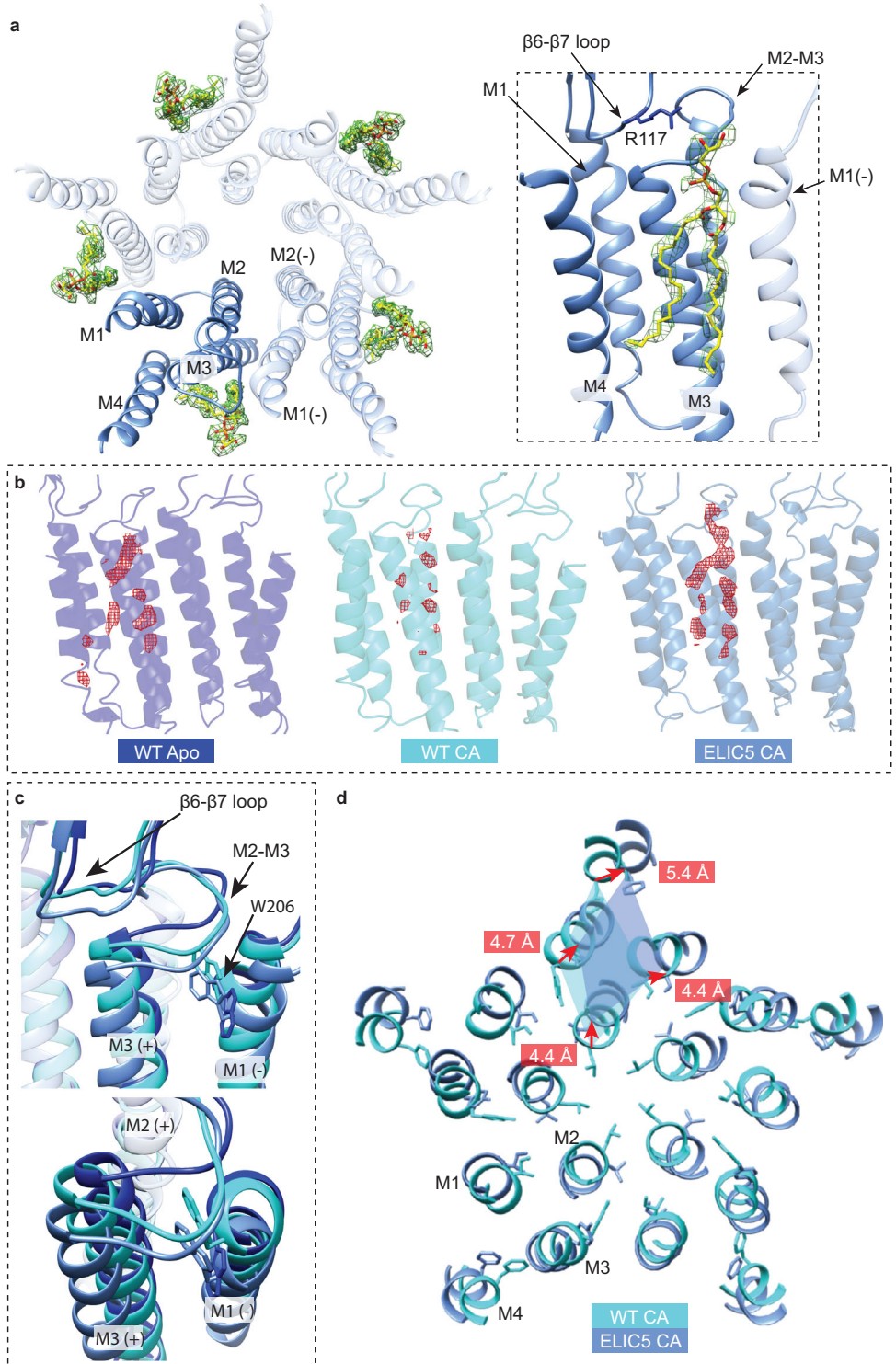

**Fig. 3 | Outer leaflet phospholipid binding site and conformational changes associated with ELIC opening. a** Left: Structure of ELIC5 TMD viewed from extracellular side showing bound POPG (yellow) in the density map (green). *Right*: Side view of ELIC5 phospholipid binding site showing POPG (yellow) in the density map (green). The R117 side chain is shown. **b** Phospholipid density (red) displayed at σ-level of 3.0 from cryo-EM density maps of WT apo, WT CA, and ELIC5 CA. **c** *Top*: Conformational changes in the β6-β7 loop and the M2-M3 linker in WT apo, WT CA and ELIC5 CA. Shown is the W206 side chain for each structure. *Bottom*: Top down view of the conformational changes of the M2-M3 linker and W206 of WT apo, WT CA and ELIC5 CA. **d** View of the TMD helices from the extracellular side at the level of 9' comparing WT CA and ELIC5 CA. Distances between C-α carbons of each structure are shown in red for each transmembrane helix. Images are from a global superposition of structures.

the diacylglycerol in an intersubunit groove formed by M4, M3 and M1 (Fig. 3a). Importantly, the mutated residues in ELIC5 are not located directly in the lipid binding site (Supplementary Fig. 6a and below). Closer examination of this site shows major conformational changes between the WT apo, WT CA and ELIC5 CA structures (Supplementary Movies 1 and 2). The β6-β7 loop progressively translates outward from WT apo to WT CA to ELIC5 CA (Fig. 3c), placing the basic residue, R117, less than 4 Å from the phospholipid headgroup in the ELIC5 CA structure (Fig. 3a, Supplementary Fig. 9b). Movement of the β6-β7 loop is associated with an outward translation of the M2-M3 linker and the top of M3 and M1, which is largest in the ELIC5 CA structure (Fig. 3c). The outward translation of these structures is linked to a global outward blooming of 4–5 Å of all four transmembrane helices leading to opening of the pore (Fig. 3d, Supplementary Movies 3 and 4). Thus, conformational changes in the phospholipid binding site of the open-channel structure of ELIC, especially in the β6-β7 loop where R117 interacts with the phospholipid headgroup, reveals a plausible mechanism for stabilization of the open-channel conformation by an anionic phospholipid such as POPG.

Activation of ELIC is also associated with movement of a semi-conserved aromatic residue, W206 in M1, which is implicated in binding of zwitterionic phospholipids[41] and located in this outer leaflet phospholipid binding site (Fig. 3c). Indeed, we observed phospholipid density directly interacting with W206 in the WT apo structures in POPC and 2:1:1 nanodiscs (Fig. 3b, Supplementary Fig. 9b). However, while W206 faces into the bilayer in the resting structure, in agonist-bound structures (WT CA and ELIC5 CA), it flips into an intersubunit space displacing M3 and the M2-M3 linker and dramatically changing the structure of this lipid binding site (Fig. 3c, Supplementary Fig. 9b). We discovered that the tryptophan fluorescence of ELIC decreases with the addition of agonist, and that this decrease is entirely attributed to W206 (no change in fluorescence is observed in W206F with and without agonist, Supplementary Fig. 10a, b). Taking advantage of this unique agonist-dependent signal, we monitored tryptophan fluorescence of ELIC after rapid mixing of propylamine, to compare the rate of W206 movement with channel opening from the thallium flux assay. Although this measurement was performed with ELIC in DDM, the rate of decrease in tryptophan fluorescence was approximately 2–4× faster than channel opening in 2:1:1 or POPC liposomes from the thallium flux assay (Supplementary Fig. 10c). This finding indicates that W206 moves away from the protein-lipid interface during activation. W206 has been reported to form cation-pi interactions with phospholipids in the resting conformation favoring zwitterionic over anionic phospholipids[42]; however, movement of W206 away from the membrane with channel activation may alter the specificity of this site for phospholipids such as PG.

The WT apo, WT CA and ELIC5 CA structures show other notable differences in the ECD and TMD associated with channel activation and opening of the pore. In the ECD, agonist binding initiates movement of loop C towards the pore vestibule clamping down on the agonist (Supplementary Fig. 11a), a conformational change previously described in ELIC and other pLGICs[32]. However, in the ELIC5 CA structure, the agonist binding site remains clamped down but translates away from the pore (Supplementary Fig. 11a). Associated with this translation, there is a general expansion of the ECD β-sheets β1-β6 (Supplementary Fig. 5a, Supplementary Movies 5–8). In the TMD, the outward blooming of the TM helices in the ELIC5 CA structure extends to the bottom of the transmembrane helices (Supplementary Fig. 11b), and is associated with an outward translation of 4.7 Å and straightening of M4 below P305, a conserved proline that produces a kink in the M4 helix in non-conducting structures of ELIC (Supplementary Fig. 11c)[26]. Interestingly, the cryo-EM density for M4 is weaker in the WT structures with an absence of density beyond I317, in contrast to ELIC5 CA, which showed a relatively stronger M4 density (Supplementary Figs. 7, 9 and 11c). Thus, ELIC opening is associated with an outward movement and

straightening of M4, while non-conducting structures show evidence of a more dynamic M4 consistent with previous reports of M4 in the desensitized state[26].

In summary, the ELIC5 CA structure shows key conformational changes associated with channel opening, and a putative mechanism by which a phospholipid stabilizes the open-channel conformation of ELIC through an outer leaflet binding site.

## POPG modulates ELIC from the outer leaflet

To further assess the possibility that POPG modulates ELIC from this binding site in the outer leaflet, we measured ELIC function in asymmetric liposomes using the fluorescence stopped-flow flux assay[13,30,43–46]. ELIC was reconstituted in 3:1 POPC:POPE liposomes, after which the sample was split and treated with methyl-β-cyclodextrin alone (no-exchange), or methyl-β-cyclodextrin with POPG (exchange) set to introduce 25 mole% POPG to the outer leaflet (Fig. 4a)[44]. The presence of outer leaflet POPG was assessed by zeta potential measurements[44]. The asymmetric proteoliposomes had a similar zeta potential as symmetric 2:1:1 POPC:POPE:POPG liposomes, indicating successful introduction of 25 mol% POPG in the outer leaflet and no significant flipping of POPG to the inner leaflet (Fig. 4b)[44,47]. Moreover, the zeta potentials of asymmetric liposomes with 25 mol% outer leaflet POPG with and without ELIC were identical, indicating that ELIC does not cause flipping of POPG from the outer to inner leaflet or, itself, alter the zeta potential (Fig. 4b).

The addition of methyl-β-cyclodextrin alone did not alter ELIC responses (Supplementary Fig. 12). However, introduction of 25 mol% POPG to the outer leaflet increased the peak agonist response approximately two-fold and reduced the rate of desensitization (Fig. 4c–e). This effect recapitulates the effect of POPG in symmetric liposomes (Fig. 1). Of note, the reconstitution efficiency and orientation of ELIC in the no-exchange and exchange liposomes are the same, since both samples are taken from the same batch of proteoliposomes. Thus, in the asymmetric liposome experiments, peak responses between no-exchange and exchange samples can be attributed to channel activity. Overall, these data show that POPG modulates ELIC from the outer leaflet, increasing peak channel activity and decreasing channel desensitization. Not having examined the effect of POPG in the inner leaflet alone, we cannot exclude the possibility that POPG can also modulate ELIC from the inner leaflet as has been previously proposed[13,26].

## POPG binds specifically to the outer leaflet site in the open-channel structure

Having established that POPG modulates ELIC gating from the outer leaflet, we further assessed whether this effect is mediated by the phospholipid bound to the outer leaflet site. We hypothesized that this site is specific for POPG and that POPG binding favors the open-channel over non-conducting conformations. To test this, we carried out streamlined alchemical free energy perturbation (SAFEP[29]) simulations to determine the relative affinities of POPC, POPE, and POPG for binding modes corresponding to the phospholipid-shaped density in WT CA and ELIC5 CA structures. Representative density of POPG bound to the ELIC5 CA structure is shown in Fig. 5a. Using SAFEP, we determined the relative probabilities of POPG, POPC, or POPE for the outer leaflet site, conditional upon the requirement that the glycerol backbone for each phospholipid remains within 6 Å of the reference configuration, which is the upper limit observed in simulations of the bound state (Supplementary Fig. 13). These calculations revealed preferential binding of POPG for the outer leaflet site in a 2:1:1 POPC:POPE:POPG membrane for both WT CA and ELIC5 CA structures (Table 1).

To investigate state dependence and the effect of membrane composition, we extrapolated to a range of ternary compositions to obtain phospholipid binding curves (Fig. 5b). In a $2_{POPC}:1_{POPE}:x_{POPG}$

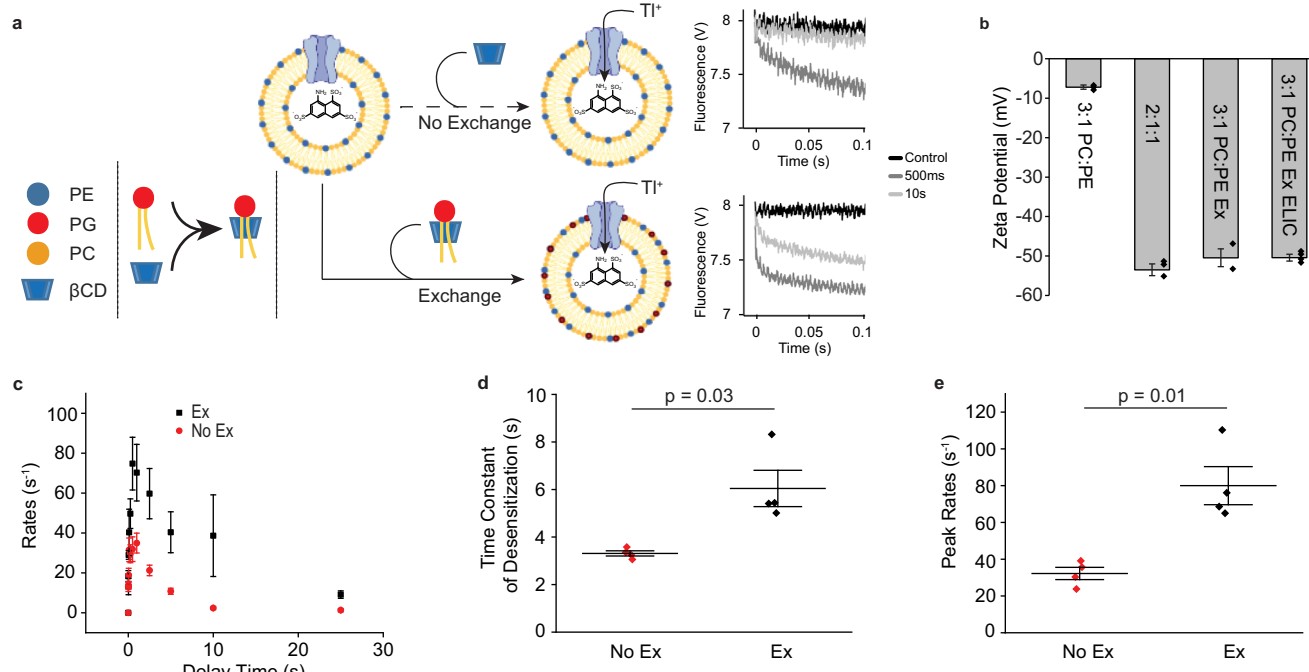

**Fig. 4 | Asymmetric liposome stopped-flow Tl⁺ flux assay. a** Schematic representation of asymmetric liposome stopped-flow Tl⁺ flux assay (created with BioRender.com). Starting with 3:1 POPC:POPE liposomes reconstituted with ELIC, the sample was either treated with mβCD (top, "No Exchange"), or treated with mβCD/POPG to introduce 25 mole% POPG to the outer leaflet (bottom, "Exchange"). *Right*: Representative fluorescence quenching traces for no exchange (top) and exchange (bottom) samples showing control (no agonist) and responses to 10 mM propylamine with 500 ms and 10 s delay times. **b** Zeta potential measurements of symmetric 3:1 POPC:POPE liposomes (3:1 PC:PE) ($n = 3$), symmetric 2:1:1 POPC:POPE:POPG liposomes (2:1:1) ($n = 3$), asymmetric 3:1 POPC:POPE liposomes treated to introduce 25 mol% POPG to the outer leaflet (3:1 PC:PE Ex) ($n = 2$), and asymmetric 3:1 POPC:POPE proteoliposomes with ELIC treated to introduce 25 mol

% POPG to the outer leaflet (3:1 PC:PE Ex ELIC) ($n = 4$). **c** Tl⁺ flux rates of WT ELIC in 3:1 PC:PE no exchange liposomes (red, "No Ex") and exchanged asymmetric liposomes where 25 mol% POPG is introduced to the outer leaflet (black, "Ex") ($n = 4$). Channel activation was elicited by 10 mM propylamine. **d** Weighted time constant of desensitization in the presence of 10 mM propylamine from Tl⁺ flux assay comparing no exchange and exchange conditions ($n = 4$). Statistical analysis was performed using a paired, two-sided T-test. **e** Peak rate of Tl+ flux in response to 10 mM propylamine in no exchange and exchange conditions ($n = 4$). Statistical analysis was performed using a paired, two-sided T-test. All data are shown as mean ± s.e. for ($n$) independent experiments. Source data are provided as a source data file.

ternary mixture, POPC rarely occupies the outer leaflet site, while POPE occupies the site at very low $x_{POPG}$. At higher $x_{POPG}$, POPG displaces POPE from the site, indicating preferential binding of POPG for the outer leaflet site. We define the $x_{50}$ as the mole fraction of POPG required to achieve 50% occupancy. From the curves in Fig. 5b, we estimate that $x_{50} \sim 10^{-5}$ and $x_{50} \sim 10^{-2}$ for the ELIC5 and WT conformations, respectively, which is consistent with POPG stabilization of the ELIC5 CA conformation in these ternary mixtures. Estimated relative stabilities of ELIC5 CA and WT CA are shown for a continuum of ternary compositions in Fig. 5c. Relative to pure POPC, where the relative stabilization is 0 by definition, adding POPE stabilizes the nonconducting WT CA conformation, and adding POPG stabilizes the open-channel ELIC5 CA conformation.

Since the lipid binding site differs between apo and agonist-bound structures especially due to movement of W206 (Supplementary Fig. 9b), we examined POPG binding to this site in the WT apo structure. Equilibrium MD simulations of POPG in the binding mode observed in the WT apo structure showed unbinding of the POPG headgroup after about 10 ns of simulation time, consistent with previous simulations in apo ELIC structures (Supplementary Fig. 14)[42]. Thus, POPG occupancy of the outer leaflet site is higher in agonist-bound structures particularly the ELIC5 CA structure.

## Functional role of the outer leaflet POPG binding site
We next tested the effect of mutations that directly interact with the phospholipid in the ELIC5 CA structure (R117, T259), or are near this site and previously shown to interact with cardiolipin (S202, R318A,

T259A and W206F) (Fig. 6a)[14]. The mutations were tested as for WT by examining the effect of 25 mol% outer leaflet POPG in paired (no exchange versus exchange) samples (Supplementary Fig. 15). Three mutations (Q264L, R117A, and T259A) showed an apparent loss-of-function in that agonist responses were low (T259A) or undetectable (Q264L and R117A); to improve the signal, we increased the amount of protein 10× for the reconstitution of these mutants. Despite the mutations having various effects on ELIC agonist responses, comparison of no exchange and exchange samples enabled assessment of the effect of POPG.

The mutations can be categorized into those where both effects of POPG (increased peak response and decreased desensitization) are abolished (R117A, S202A), those where POPG modulation is preserved (Q264L), and those where POPG only decreases desensitization (R318A, T259A, W206F) (Fig. 6b–d). R117A, which mutates a basic residue near the phospholipid headgroup in the open-channel structure (Fig. 6a), strikingly ablates both measures of POPG modulation (Fig. 6c, d). S202A at the top of M1 also produced a loss of POPG modulation. Q264L activation and desensitization was very fast such that maximum activity was seen at the shortest time point (i.e., 10 ms delay time), making the exact peak response and rate of desensitization uncertain. Nevertheless, Q264L showed a 2× increase in the measurable peak response with POPG (Fig. 6d) and significantly higher activity at delay times of 2.5 to 10 s with POPG (Fig. 6b), indicating that POPG positively modulates peak response and desensitization in this mutant. Consistent with this finding, Q264 is > 7 Å removed from the phospholipid headgroup in the WT CA and ELIC5 CA structures (Fig. 6a

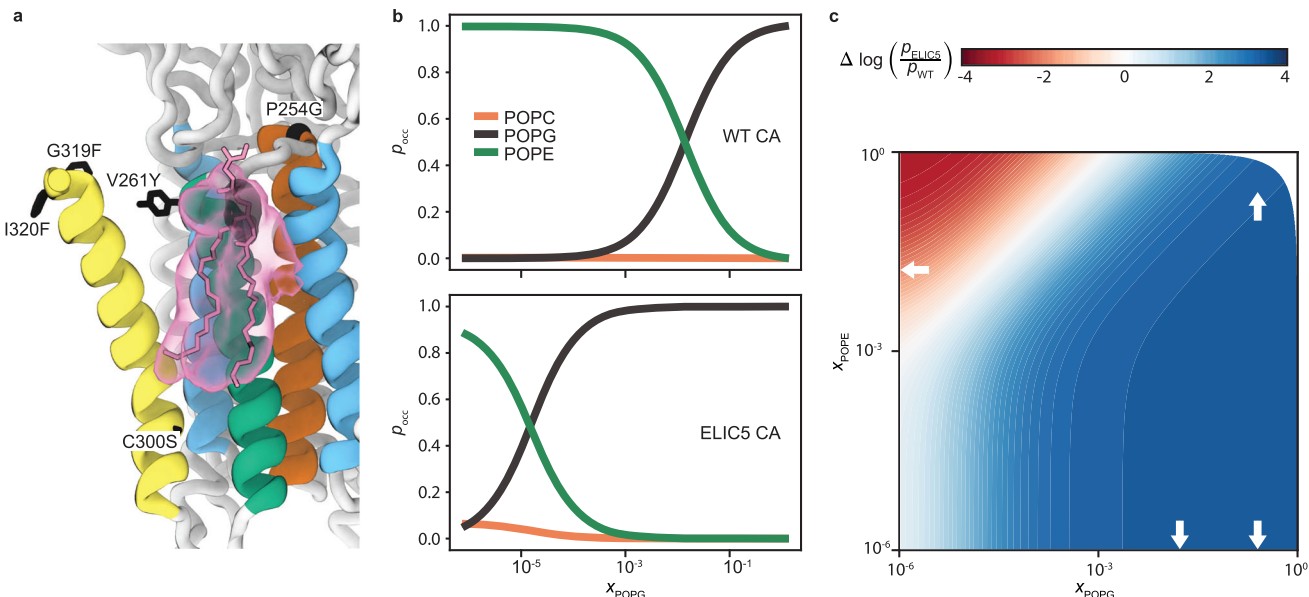

**Fig. 5 | Predicted lipid occupancy for the outer leaflet phospholipid binding site. a** ELIC5 CA structure showing lipid density from a brief 20 ns equilibrium MD simulation to estimate expected thermal fluctuations of the glycerol backbone around the structurally-modeled POPG (pink licorice). The region of glycerol backbone density is shown in black, while the remaining lipid density is shown in pink. The five mutations of ELIC5 are shown in black licorice. **b** The SAFEP-calculated probability ($p_{occ}$) that the structurally-identified site will be occupied by each of three possible lipids in a $2_{POPC}:1_{POPE}:x_{POPG}$ mixture, for both the WT CA

conformation (top) and the ELIC5 CA conformation (bottom). **c** The predicted relative conformational stability $\Delta \log\left(\frac{p_{ELIC5}}{p_{WT}}\right)$ as a function of mole fraction of POPG and POPE in a POPG:POPE:POPC mixture. Red and blue correspond to greater stability of the WT CA and ELIC5 CA conformations, respectively. White arrows indicate compositions where bulk calculations were carried out; remaining values were extrapolated as described in Methods. Data in panels b and c represent over 3 µs of simulation; pharmacological models relied on SAFEP-calculated parameters as described in Methods.

and Supplementary Fig. 9b). R318A, T259A and W206F abolished the effect of POPG on peak response, but still showed an effect of POPG on desensitization, with some variability (Fig. 6b–d). The reason for this discrepancy, especially T259A which showed a profound slowing of desensitization with POPG, is unclear (Fig. 6c). While the mutations have complex effects on channel function and POPG modulation, the apparent loss of POPG modulation in R117A, which disrupts a charged interaction with the POPG headgroup, and the loss of POPG effect on peak response in multiple mutations implicates the outer leaflet intersubunit site in the mechanism of POPG modulation.

## Discussion

It has been known for four decades that anionic phospholipids are critical for maximal agonist response in the nAChR[8,9]. More recently, anionic phospholipids have been shown to modulate ELIC[13,14]. Many structures of different pLGICs including ELIC show bound phospholipids[19,23–25], intimating that phospholipid modulation is mediated by direct binding to specific site(s). By capturing an open-channel conformation of ELIC, the results of this study suggest a mechanism for POPG modulation of ELIC through an outer leaflet intersubunit site. While the presence of a stronger phospholipid density in the open-channel structure relative to non-conducting

structures does not exclude the possibility of POPG interaction at this site in non-conducting conformations, it does suggest higher occupancy of POPG in the open-channel conformation, which was substantiated by the FEP calculations. One caveat to this interpretation is the possibility that the ELIC5 mutations promote interaction of POPG at this site. While we cannot rule out this possibility, it seems less likely since the mutated residues do not make direct contact with the phospholipid. A recent structure of ELIC in SMA nanodiscs showed cardiolipin bound to an overlapping outer leaflet site[14]. Like POPG, cardiolipin also positively modulates ELIC. Another study using MD simulations showed that zwitterionic lipids such as PC selectively bind to this site forming a cation-pi interaction with W206[41]. However, both studies examined resting state structures where W206 faces into the bilayer. In our agonist-bound structures including the open-channel structure, W206 has turned into an intersubunit space, drastically changing the structure of this lipid binding site and making PG occupancy more favorable. Therefore, it seems plausible that cardiolipin occupancy of this site may also differ in open-channel relative to non-conducting conformations[14]. Interestingly, a recent study demonstrated that the polyunsaturated fatty acid, docosahexaenoic acid (DHA), inhibits ELIC by binding to a site overlapping with the POPG site. DHA produces the opposite effect as POPG: it decreases peak agonist response and increases desensitization[12]. Thus, it appears that competition by different lipids (i.e., anionic phospholipids and fatty acids) for this site modulates the relative stability of open-channel and non-conducting conformations in ELIC.

While we propose a mechanism by which POPG stabilizes the open-channel conformation of ELIC, it is also possible that POPG and POPE increase peak responses by promoting a coupled receptor. Previous studies in the nAChR showed that in the absence of anionic phospholipids, the receptor adopts an uncoupled conformation[11,48]. However, the structures of ELIC in POPC and 2:1:1 nanodiscs from the present study show no differences that point to coupled and uncoupled conformations. Furthermore, fourier transform infrared spectroscopy

## Table 1 | Free energy differences calculated from SAFEP

| | Local Environment | POPE to POPC (kcal/mol) | POPE to POPG (kcal/mol) |
|---|---|---|---|
| Transformation of Bulk Lipid | $2_{POPC}:1_{POPE}:1_{POPG}$ | $1 \pm 1$ | $-17 \pm 2$ |
| | $3_{POPC}:1_{POPG}$ | $-1 \pm 1$ | $-21 \pm 3$ |
| | POPC | $-1 \pm 1$ | $-17 \pm 1$ |
| Transformation of Bound Lipid | WT CA | $5 \pm 1$ | $-19 \pm 2$ |
| | ELIC5 CA | $3 \pm 1$ | $-24 \pm 2$ |

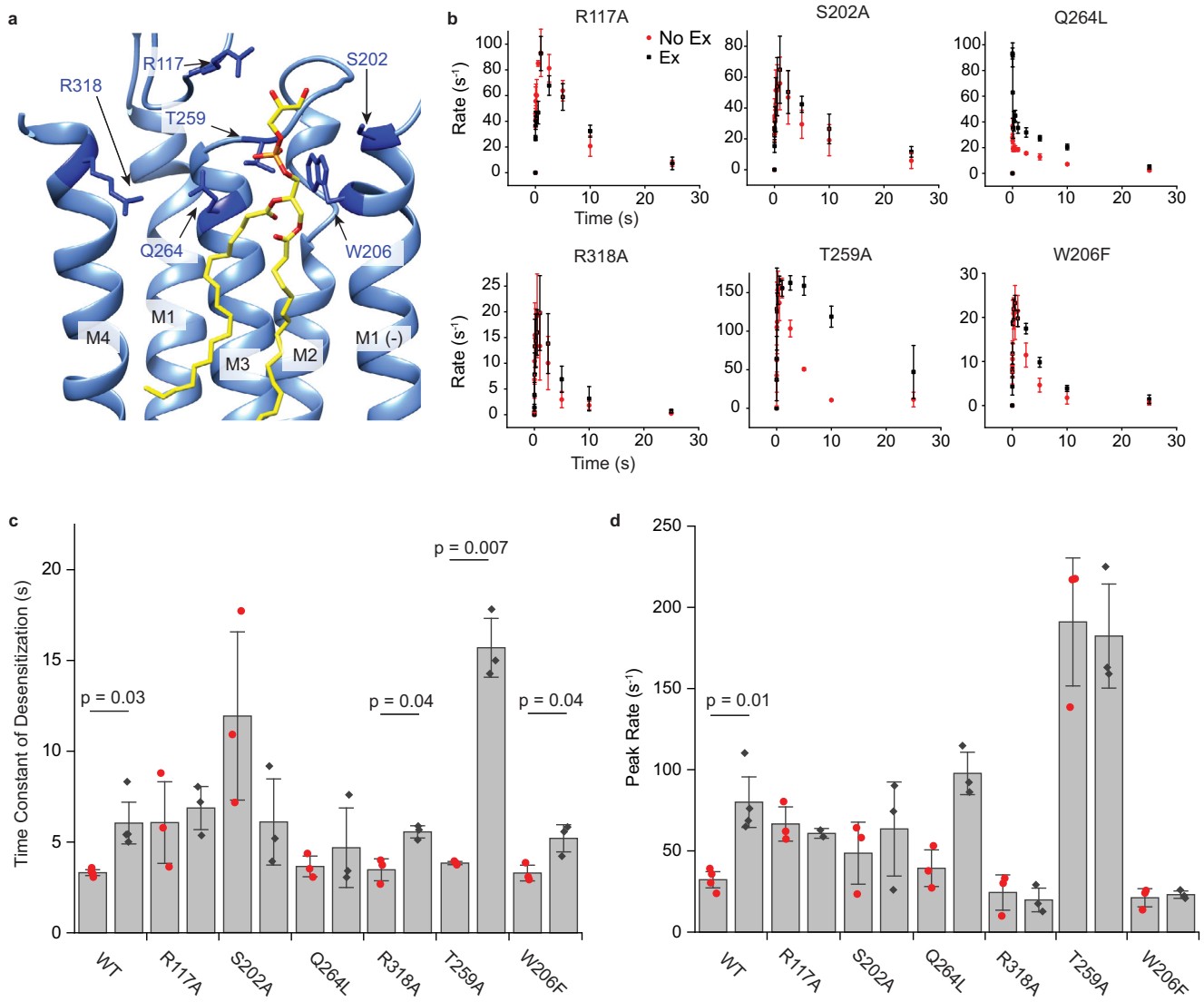

**Fig. 6 | Mutational analysis of the outer leaflet phospholipid binding site.**
**a** ELIC5 CA structure showing POPG (yellow) at the outer leaflet site and residues (dark blue) targeted by mutagenesis. **b** Tl$^+$ flux rates of mutants in response to 10 mM propylamine for no-exchange (red) and exchange (black, 25 mol% POPG in the outer leaflet) conditions ($n$ = 3). Source data are provided as a source data file. **c** Weighted time constant of desensitization in response to 10 mM propylamine for no-exchange (red) and exchange (black) conditions ($n$ = 4 for WT, $n$ = 3 for all mutants). **d** Same as **c** showing peak rates of Tl$^+$ flux in response to 10 mM propylamine for no-exchange (red) and exchange (black) conditions ($n$ = 4 for WT, $n$ = 3 for all mutants). All data are shown as mean ± s.e. for ($n$) independent experiments. Statistical analysis was performed using a paired, two-sided T-test.

measurements indicate that ELIC does not adopt an uncoupled conformation with the same features as the nAchR in POPC membranes[33]. Therefore, the current data do not support such a mechanism in ELIC.

Using asymmetric liposomes, we establish that POPG modulates ELIC from the outer leaflet. The observed effect is unlikely to result from flipped inner leaflet POPG since the zeta potentials of asymmetric liposome samples were not significantly different from the 25% POPG symmetric liposome control, deviating no more than 3 mV. This means that a negligible amount of POPG could be present in the inner leaflet with an estimated upper limit of ~3–4%[44]. In contrast, the effect of 25% outer leaflet POPG, especially on desensitization, mimics what is observed in 2:1:1 (25% POPG) symmetric liposomes supporting the conclusion that POPG acts from the outer leaflet. It is also possible that methyl-β-cyclodextrin could impact flux rates by altering liposome size or integrity; this seems unlikely given that methyl-β-cyclodextrin alone did not significantly impact ELIC-mediated flux rates or produce a noticeable loss of intra-liposomal ANTS. POPG modulation from the outer leaflet agrees with the finding that PG is enriched in the outer

leaflet of gram negative bacteria membranes where ELIC is expressed[49]. Interestingly, lipid densities have been observed in the structures of the muscle nAchR, GluCl and the 5-HT$_{3A}$R at this outer leaflet site[19,23,28,50]. In addition, MD simulations of the glycine receptor showed state-dependent binding of phospholipids between inactive and active conformations at the top of M3 and M1—a site analogous to the POPG binding site in ELIC[51]. Interestingly, R117 in the β6-β7 loop, which forms an electrostatic interaction with the PG headgroup in the open-channel structure, is conserved in GLIC and human pLGICs (Supplementary Fig. 16). Thus, there is structural and computational evidence to suggest that a mechanism, similar to that proposed in the present study, exists in human pLGICs. Anionic phospholipids are localized to the inner leaflet of mammalian plasma membranes[52]; yet, it is thought that some anionic phospholipids such as phosphatidylserine exist in the outer leaflet of neuronal membranes and are dynamically regulated[53]. It is possible that anionic phospholipids in the outer leaflet fine-tune agonist responses of mammalian pLGICs through this site, or that other lipids more abundant in the outer leaflet act through this site.

We report an open-channel structure of ELIC; all prior structures of ELIC show a resting conformation[14,26,32,54,55], with the exception of an agonist-bound structure in POPC nanodiscs displaying partial activation[32]. While we cannot exclude the possibility that the ELIC5 structure is an artifact of mutations, the fact that this mutant has a high steady-state open probability in the presence of agonist shows excellent agreement between structure and function. The changes transitioning from resting to activated conformations in ELIC are generally conserved in other pLGICs, although our structures reveal marked differences between non-conducting and open-channel conformations that are not appreciated to the same extent in other pLGICs structures[19,23,36,56,57]. These include a significant outward blooming of the TM helices and expansion of the ECD β-sheets (β1-β6), indicating that the transition from pre-active to open involves global conformational changes spanning the ECD and TMD. Such an observation is consistent with a model that long-range global structural rearrangements underlie the transitions between functional states in pLGICs[58].

## Methods

### Protein mutagenesis, expression, and purification

The pET-26-MBP-ELIC was provided by Raimund Dutzler (Addgene plasmid # 39239) and was used for the generation of both WT and mutant ELIC. Site-directed mutagenesis was conducted using the Quikchange method and subsequently confirmed by Sanger sequencing (Genewiz). WT and mutant ELIC were expressed as previously reported[15,54] in OverExpress C43 (DE3) *E. Coli* (Lucigen). Terrific Broth (Sigma) was used to grow cultures which were induced with 0.1 mM IPTG for ~16 h at 18 °C. The cells were pelleted, resuspended in Buffer A (20 mM Tris pH 7.5, 100 mM NaCl) with complete EDTA-free protease inhibitor (Roche), and lysed using an Avestin C5 emulsifier at ~15,000 psi. Membranes were pelleted by ultracentrifugation, resuspended in Buffer A, solubilized with 1% DDM (Anatrace), and incubated with amylose resin (New England Biolabs) for 2 h. The amylose resin was washed with 20 bed volumes of Buffer A containing 0.02% DDM and 0.05 mM TCEP (ThermoFisher Scientific), and eluted with 40 mM maltose (Sigma). The eluted protein was digested with HRV-3C protease (ThermoFisher Scientific) (10 units per mg ELIC) overnight at 4 °C then purified over a Sephadex 200 Increase 10/300 (GE Healthcare Life Sciences) size exclusion column in Buffer A with 0.02% DDM.

### Nanodisc reconstitution

Phospholipids (POPC, POPE, POPG) (Avanti Polar Lipid) were used as received or combined to the desired ratio in chloroform, then dried under a stream of nitrogen and placed in a vacuum desiccator overnight. The lipids were then rehydrated in Buffer A to a concentration of 7.5 mg/ml (~10 mM), sonicated for 5–10 min, and subjected to 5-10 freeze-thaw cycles using liquid nitrogen and a 37 °C heat block, and finally extruded with a 400 nm filter 40× (Avanti Polar Lipids). Samples were prepared using a 1:2:200 ratio of ELIC:MSP1E3D1:phospholipid. 300 µl samples were prepared by combining 74 µL Buffer A, 30 µL of ~10 mM liposomes, and 70 µL of 1% DDM in Buffer A yielding a 0.23% DDM solution which was incubated at RT for 3 h. Next, 93 µL of 3 mg/mL (~16 µM) ELIC, or ELIC mutant, was added with 33 µL of MSP1E3D1 at 3 mg/mL, then incubated at RT for 1.5 h. MSP1E3D1 was expressed and purified according to previous reports, without removal of the His-tag[59]. The pMSP1E3D1 plasmid was a gift from Stephen Sligar (Addgene plasmid # 20066). Following incubation, the samples were cooled on ice for 10 min followed by the addition of 80 mg Bio-beads SM-2 Resin (Bio-Rad), then rotated at 4 °C overnight. The nanodisc solution was purified on a Sephadex 200 Increase 10/300 column using 10 mM HEPES with 100 mM NaCl at pH 7.5.

### Cryo-EM sample preparation and imaging

3.0 µl of purified ELIC nanodiscs at an approximately concentration of 1.2 mg/ml were pipetted onto Quantifoil R2/2 copper grids post plasma cleaning in an $H_2/O_2$ plasma for 60 s (Solaris 950. Gatan). Agonist bound samples were prepared by combining the purified channels with cysteamine to a final agonist concentration of 10 mM. Post application of the sample, the grid was blotted for 2 s at -100% humidity and vitrified via plunge freezing into a solution of liquid ethane using a Vitrobot Mark IV (ThermoFisher Scientific). Post vitrification, grids were mounted in Autogrid holders and transferred to a Titan Krios 300 kV Cryo-EM (ThermoFisher Scientific). Datasets were collected in an automated fashion with the EPU acquisition software (EPU 2.9.0.1519) using a Gatan K2 Summit camera mounted on a GIF BioQuantum 968 energy filter (Gatan). Movies were collected in counting mode with a nominal pixel size of 1.1 Å with the GIF set to a slit width of 20 eV operating in zero-loss mode. Dose fractionated movies were acquired over a defocus range of −1 to −2.5 µm. Each movie consisted of 40 individual frames with a per-frame exposure time of 200 ms, resulting in a dose of 66 electrons per Å$^2$.

For cryo-EM of liposomes, 3.0 µl of purified liposomes (POPC or 2:1:1) at 5 mg ml$^{-1}$ were pipetted onto Lacey carbon grids which were glow discharged for 12 seconds at 15 mA using a Gloqube (EMS). Post application of the sample, each grid was blotted for 2 seconds at -100% humidity and vitrified via plunge freezing into a solution of liquid ethane using a Vitrobot Mark IV operating at 22 °C (ThermoFisher Scientific). Post vitrification, grids were mounted in AutoGrid holders and transferred to a Glacios 200 kV Cryo-EM (ThermoFisher Scientific). Datasets were collected in an automated fashion with the EPU acquisition software using a Thermo Fisher Scientific Falcon 4 (Thermo-Fisher Scientific). Micrographs were collected with a nominal pixel size of 1.9 Å. For cryo-EM images of liposome samples, liposome diameter was manually determined in ImageJ using the polygon selection tool.

### Single particle analysis and model building

Raw movies were motion corrected and subjected to dose weighting with MotionCor2[60]. Motion corrected and dose weighted images were subject to contrast transfer function (CTF) determination using GCTF v1.06[61]. After CTF determination, low-quality micrographs were manually discarded. For each dataset ~2500 particles were hand selected for 2D class averaging from which templates were derived for automated particle picking in RELION3.1[62]. Particles were extracted and subjected to multiple rounds of 2D and 3D classification. For WT ELIC CA in POPC nanodiscs, an initial 3D model was generated using a 40 Å-low-pass filtered map of PBD 2YN6 [https://www.wwpdb.org/pdb?id=pdb_00002yn6] and used for subsequent 3D classifications. The other five datasets utilized a 40 Å-low-pass filtered map from the WT ELIC CA POPC dataset as an initial model. 3D classification was performed using C5 symmetry and no significant differences were identified among good classes. The best 3D refined maps were then masked and post-processed, followed by CTF refinement and Bayesian polishing. To compare the strength of the phospholipid density between WT apo, WT CA and ELIC5 CA structures, each structure was low-pass filtered to 3.5 Å[63]. Next, the lipid density was visualized in PyMOL 2.5.2 by adjusting the σ-level to 3.0[64]. Other images of the structures were generated using UCSF Chimera 1.14[65].

An initial model of the WT CA 2:1:1 structure was generated by docking an ELIC crystal structure (PDB 2YN6 [https://www.wwpdb.org/pdb?id=pdb_00002yn6]) into the cryo-EM density map using UCSF Chimera and then performing real space refinement using PHENIX 1.19.2-4158[65,66]. The remaining structure was manually built de novo into the density map using COOT 0.9.8.1 prior to iterative real space refinement using PHENIX[67]. Prior to finalizing the model manual adjustments were made using COOT and verified in PHENIX. The other structures were built by docking the WT CA 2:1:1 structure into each density map and following the same procedure above. A non-proteinaceous density was observed in the agonist binding site of structures obtained in the presence of 10 mM cysteamine. Cysteamine was placed in this density in an orientation previously shown to be

favorable by MD simulations[32]. Resolved phospholipid-like densities (a head group and two branching tails) were fit with POPG for those structures determined in 2:1:1 nanodiscs and POPC for those structures determined in POPC nanodiscs. Phospholipids were generated using ELBOW in PHENIX.

## Giant liposome excised patch-clamp recordings of ELIC

Giant liposome formation and excised patch-clamp recordings of ELIC were performed as previously described[12,13]. 0.5 mg of WT ELIC, ELIC3 or ELIC5 were reconstituted in 5 mg of 2:1:1 POPC:POPE:POPG liposomes, using DDM destabilization and biobeads. 10 mM MOPS pH 7 and 150 mM NaCl (MOPS buffer) was used throughout. Proteoliposomes were pelleted and resuspended in ~80 μl, and frozen in 10 μl aliquots. Giant liposomes were formed by dehydrating 10 μl of proteoliposomes on a glass coverslip in a vacuum desiccator for ~20 min followed by rehydration with 100 μl of MOPS buffer overnight at 4 °C and 2 hr at RT the following day. The giant liposomes were resuspended by pipetting and applied to a recording chamber. The liposomes were allowed to settle for 15 min, after which free liposomes and debris were removed by exchanging the bath solution. Bath and pipette solutions contained MOPS buffer and 0.5 mM BaCl$_2$ (Sigma). The addition of BaCl$_2$ was necessary to facilitate the formation of giga-ohm seals. However, this concentration of BaCl$_2$ also leads to a right-shift in the agonist dose-response curve, such that the EC$_{50}$ for cysteamine (Sigma) is ~4-5 mM instead of 1-2 mM. Of note, no BaCl$_2$ was used in the fluorescence stopped-flow liposomal flux assay, such that the dose response curve for propylamine (Sigma) in 2:1:1 liposomes yielded an expected EC$_{50}$ of ~2 mM. All recordings were voltage-clamped at −60 mV and data collected at 5 kHz using an Axopatch 200B amplifier and Digidata 1440 A (Molecular Devices). Rapid application or washout of agonist were performed using a three-barreled flowpipe mounted to and controlled by a SF-77B fast perfusion system (Warner Instrument Corporation). The 10−90% exchange time was previously determined to be ~2 ms in this system using a liquid junction current across an open pipette, such that solution exchanges are sufficiently fast to capture ELIC activation and deactivation kinetics. Deactivation time courses were fit with a single exponential while the time course of desensitization was fit with a double exponential. Weighted time constants were calculated using Eq. 1:

$$\text{Weighted Tau} = \frac{(A1 \times \tau1) + (A2 \times \tau2)}{A1 + A2} \tag{1}$$

where A1 and A2 are the amplitudes of the first and second exponential components. The data were collected and analyzed using pClamp10.4 software.

## Stopped-flow fluorescence recordings

The fluorescence sequential-mixing stopped-flow liposomal flux measurements were carried using an SX20 stopped-flow spectrofluorometer (Applied Photophysics) and Pro-Data SX software[13,30]. 7.5 mg of the respective lipid mixture (POPC, 1:1 POPC: POPE, 1:1 POPC:POPG, 3:1 POPC:POPE, 3:1 POPC:POPG, 2:1:1 POPC:POPE:POPG) was dried in a glass vial to a thin film under a constant stream of N$_2$. The lipids were then further dried under vacuum in a desiccator overnight. The following day the lipids were hydrated in 500 μl of reconstitution buffer (100 mM NaNO$_3$, 10 mM HEPES, pH 7) (Buffer R), 250 μl of 75 mM 8-Aminonaphthalene-1,3,6-Trisulfonic Acid (ANTS, ThermoFisher Scientific) in buffer R, and doped with ~18 mg of CHAPS (Anatrace). The mixture was vortexed and then sonicated with heat until completely solubilized. Following the sonication, the solution was brought to RT then 3.5 μl of 6.7 mg/ml ELIC in Buffer A with 0.02% DDM was added (3 μg per mg of lipid). This combined solution was incubated at RT for 30 min, followed by the addition of 1 ml of 1:1 v/v SM-2 BioBeads in Buffer R and an additional 250 μl of

75 mM ANTS in Buffer R. This mixture was incubated at RT for 2.5 hr with gentle mixing. The liposome suspension was then extruded with an Avanti mini-extruder and 0.1 μm membrane (30×). The extruded liposomes were stored at 13 °C overnight with gentle rotation. The next day, the liposomes were diluted to 2.5 mL using assay buffer (140 mM NaNO$_3$, 10 mM HEPES, pH 7) (Stopped-flow Buffer A), loaded into a PD-10 desalting column (Cytiva), eluted with 3 mL of Stopped-flow Buffer A and subsequently diluted 5-fold with Stopped-flow Buffer A.

Using propylamine as the agonist, the assay and analysis was carried out as previously described[13,30]. ELIC-containing liposomes were first mixed 1:1 with Buffer A containing 2× the concentration of propylamine. After a variable delay of 10 ms to 25 s, the sample was then mixed 1:1 with a quenching buffer (90 mM NaNO$_3$, 50 mM TlNO$_3$, 10 mM HEPES, pH 7), and ANTS fluorescence (ex. 360 nm, fluorescence above 420 nm) was recorded for 1 s. For each delay time, 5 repeat measurements were obtained. The data were collected using Applied Photophysics Pro-Data SX software. The fluorescence quenching traces were fit to a stretched exponential (Eq. 2a) and the rate of Tl$^+$ influx was determined at 2 ms (Eq. 2b).

$$F_t = F_\infty + (F_0 - F_\infty) \cdot e^{\left\{-\left(\frac{t}{\tau}\right)^\beta\right\}} \tag{2a}$$

$$k_t = \left(\frac{\beta}{\tau}\right) \cdot \left(\frac{2ms}{\tau}\right)^{(\beta-1)} \tag{2b}$$

where $F_t$, $F_\infty$, $F_0$ are the fluorescence at time t, final fluorescence and initial fluorescence. τ is the time constant, β the stretched exponential factor, and $k_t$ the calculated rate of Tl$^+$ influx at 2 ms. The time course of desensitization was also fit to a double exponential and a weighted time constant was calculated as was done for the patch-clamp data using Origin 2019b. All replicates were taken from independent samples.

## Tryptophan stopped-flow fluorescence measurements

To measure the tryptophan (Trp) fluorescence emission of ELIC, WT or W206F ELIC was diluted to 3 ml (0.02 mg/ml) and an emission spectrum was collected in a quartz fluorescence cell using a Fluoromax Plus Spectrofluorometer (Horiba Scientific). Excitation was set at 295 nm and an emission spectrum was collected from 320 to 380 nm. Trp fluorescence measurements with rapid mixing of agonist was carried out using the SX20 stopped-flow spectrofluorometer, outfitted with a 295 nm LED excitation source. Purified ELIC was diluted to 0.1 mg/ml in Trp Stopped-flow Buffer A (10 mM Tris pH 7.5, 100 mM NaCl, 0.05% DDM). A single mixing experiment was performed by mixing ELIC with equal volumes of Buffer A + propylamine. Propylamine was prepared at varying concentrations at 2x the final concentration. The instrument dead time was ~1 ms, enabling capture of most of the time course of change in Trp fluorescence. The time course of decrease in Trp fluorescence after rapid mixing with propylamine was fit to a single exponential to obtain a time constant for the rate of decrease and a measure of the extent of decrease. All replicates were taken from distinct samples.

## Asymmetric liposome formation and analysis

Acceptor liposomes were prepared as described in *the Stopped-flow fluorescence recordings* section until completion of the extrusion step. The preparation was scaled up 2x to produce samples for the no-exchange and exchange conditions. After extrusion and storage of the liposomes overnight at 13 °C, the double liposome preparation was split into two equal volume aliquots of ~1.5 ml. Using the excel spreadsheet provided by Markones et al.[44], donor POPG and methyl-β-cyclodextrin (MβCD, Sigma) concentrations and volumes were calculated to produce an outer leaflet POPG content of 25 mol% in the

liposomes. The calculations assumed a ~6.66 mM lipid concentration and 1.5 mL volume of the acceptor liposome solution[44]. A MβCD-POPG (50% saturation of MβCD by POPG) solution was formed by combining 224 μL 7.5 mM POPG in Stopped-flow Buffer A, 650 μl 300 mM MβCD in Stopped-flow Buffer A, and 126 μL Stopped-flow Buffer A, which was then mixed at 50 °C and 1000 RPM for 20 min using an Eppendorf thermomixer. In parallel, an analogous POPG-free solution was generated and treated in the same manner except 224 μL of Stopped-flow Buffer A was used in lieu of the POPG solution. Next, the solutions and thermomixer were cooled to 28 °C. Following cooling, the 1.5 ml of acceptor liposomes were combined with the respective 1 ml MβCD solution, either containing POPG (exchange condition) or no POPG (no exchange condition). These samples were mixed at 28 °C and 400 RPM for 20 min using a thermomixer to complete the transfer of POPG to the liposomes. After mixing, the ~2.5 ml of each respective sample were loaded on a PD-10 column and eluted with 3 mL Stopped-flow Buffer A. The fluorescence stopped-flow liposomal flux assay and analysis of the data were carried out on both the no exchange and exchange samples in sequence as described in *Stopped-flow fluorescence recordings*[13]. All replicates were taken from distinct samples.

## Zeta potential measurements

The zeta potential (ζ) is the electrical potential on the surface of nanoparticles (e.g., liposomes), specifically at the slipping plane, which is between a diffuse layer of solvent and ions interacting with the particle surface and the mobile bulk solvent. For liposomes, this electrical potential is determined by the overall charge on the outer leaflet of the lipid membrane, and can be used to assess introduction of POPG to the outer leaflet of liposomes[44]. ζ measurements were made on a Malvern Zetasizer Nano-ZS ZEN 3600 (Zetasizer nano software v3.30) using a flow-through high concentration zeta potential cell (HCC) following a previously described method[44,45]. The instrument was equilibrated for 20 min at 28 °C prior to analysis. Instrument settings included: temperature 28°, 90 s incubation time, 10-100 runs, and the dispersant settings were set as water for all samples. ζ was measured in triplicate for each sample. Prior to each measurement the HCC was flushed with copious volumes of preheated ddH$_2$O water and dried under a flow of N$_2$. The sample was prepared using liposomes immediately after desalting on the PD-10 column, and diluting 100 μl of liposomes with 100 mM HEPES at pH 7 to a final volume of 1 ml (final MβCD ~ 4.3 mM). The sample was then loaded into a 1 ml syringe and the liposome suspension was driven into the cell until all air was displaced. Upon each subsequent measurement an additional ~250 μl was pushed through the cell to displace old solution. After use, the HCC was disassembled and cleaned with 1% Hellmanex II in ddH$_2$O and dried[44].

## Molecular dynamics simulation

Initial protein–lipid complexes were modeled directly from cryo-EM maps with lipid densities. The first 11 protein residues were not modeled. The lipid densities were modeled using POPC, POPE, or POPG. These initial structures were then inserted into POPC membranes and equilibrated using default procedures from CHARMM-GUI[68]. All simulations were carried out using the CHARMM36m forcefield[69] with cation-pi corrections[70] Simulations were executed using NAMD 2.14[71] in a semi-isotropic NPT ensemble tailored for membranes: a Langevin piston barostat with a target temperature of 303.15 K and a target pressure of 1 atmosphere with isotropic fluctuations enforced in the membrane plane. The piston period was increased to 200 steps (2 fs time step) and the decay to 100; this reduced instabilities due to higher frequency fluctuations. Following the default relaxation scheme of CHARMM-GUI[68,72], systems were further relaxed for at least 20 more nanoseconds to ensure equilibration and to determine the expected deviation of the glycerol backbone due to thermal fluctuations around the modeled binding pose. As described below, these simulations are

essential for parameterizing the Distance-to-Bound-Configuration (DBC) coordinate used in the SAFEP approach, but are not intended to thoroughly sample the full range of configurations available to the lipid or protein. Membrane-only systems were similarly created and equilibrated.

## Streamlined alchemical free energy perturbation (SAFEP) simulations

We performed binding free energy estimation using the molecular dynamics simulation settings previously described, with alchemical calculations using the SAFEP framework[29]. SAFEP is a simplified approach to calculating free energies of ligand binding using free energy perturbation (FEP). A primary challenge in FEP calculations is preventing spontaneous unbinding during simulation which would adversely impact the sampling efficiency. In the SAFEP framework[29], the bound state is defined as a region of the configuration space for the ligand (RMSD) relative to the bound pose in the protein frame of reference (a.k.a. the "Distance-to-Bound-Configuration" or DBC). Interleaved-double wide sampling (IDWS) improves efficiency of any FEP by sampling in both the forward and backward lambda directions alternately (unlike traditional FEP in which each lambda value is simulated twice; once with forward sampling and once with backward sampling).

As in other relative FEP methods, the zero-sum nature of thermodynamic cycles is exploited to calculate interesting relative free energy differences from more computationally accessible data. In this case, the alchemical free energies of transformation calculated by SAFEP (summarized in Table 1) can be combined to calculate relative free energies of replacement of one lipid with another:

$$\Delta\Delta G_{Bulk \to M}^{a \to b} = \Delta G_{Bulk \to M}^{b} - \Delta G_{Bulk \to M}^{a} = \Delta G_{M}^{a \to b} - \Delta G_{Bulk}^{a \to b} \quad (3)$$

where $a$ and $b$ are generic lipid species, M is a protein conformation, $\Delta G_{M}^{a \to b}$ is the free energy cost of transforming a lipid of species $a$ to species $b$ bound to M, $\Delta G_{Bulk \to M}^{b}$ is the free energy cost of transferring a lipid of species $b$ from the bulk to a protein in state M, and $\Delta\Delta G_{Bulk \to M}^{a \to b}$ is the relative free energy of exchanging a bound lipid of species $a$ with a lipid of species $b$ from the bulk.

Free energies of transformation of bound ligands ($\Delta G_{M}^{a \to b}$) were calculated for each of two conformations (WT and ELIC5) and each of three $a$-$b$ pairs (POPC vs POPE, POPG vs POPE, POPC vs POPG). In this case, the bound pose could be defined using a relaxed conformation of the initial protein-lipid complex. The DBC only included the glycerol atoms of the lipid since inclusion of the more flexible acyl chains and headgroups would require a much looser restraint, thus reducing the sampling benefits.

Following equilibration of the membrane-embedded proteins, the bound lipid was replaced by one of three hybrid (or "dual") topologies: alchemical lipids which share acyl chains and glycerol cores, but bear two separate phospholipid headgroups. Protein-lipid FEP calculations were carried out with half flat-bottom DBC restraints on the bound lipid to prevent unbinding. The upper wall was placed near the 95$^{th}$ percentile of the unbiased simulation of POPC bound to ELIC5 (the most weakly bound protein-lipid complex), which corresponded to 6 Å (Supplementary Fig. 13). 120 lambda windows of 3 ns per window were run in an embarrassingly parallel arrangement. The systems were equilibrated at each lambda value for at least one nanosecond prior to collection of FEP data. All SAFEP calculations used IDWS for simultaneous calculation of the reverse transformation. Due to the computational cost of these simulations, a single replica was run for each lipid transformation for each conformation (6 individual calculations total).

Membrane-only simulations used to calculate $\Delta G_{Bulk}^{a \to b}$ lacked DBC restraints but required two modifications to prevent lipid flipping during FEP. The electrostatic scaling was started at lambda = 0.25 instead of 0.5; this setting prevented complete discharging of the lipid.

Forty linearly spaced 1.4 ns FEP windows were used, with 0.5 ns of equilibration prior to collection of FEP data. This modification was not sufficient to prevent lipid flipping for some transformations involving POPE and POPG (Supplementary Figs. 17 and 18), and in these cases a flat-bottom harmonic restraint was placed between the central carbon on the FEP lipid and a lipid in the opposite leaflet (lower bound 28.5 Å; upper bound 41.5 Å; force constant 4.0 kcal/mol/Å$^2$). Five replicas were run for each lipid comparison in each membrane composition. Replicas were generated by selecting a different lipid for each calculation.

## SAFEP analysis and convergence-testing

The output from the FEP calculations was used to estimate the difference in free energy between the two end states using an accumulated Bennett's Acceptance Ratio estimator (BAR) as implemented in PyMBAR[73] using the alchemlyb interface[74], and custom scripts. Decorrelated samples were obtained using subsampling utilities available in those libraries. Replicate data were aggregated for final free energy estimations. Errors were estimated by using the BAR estimator as implemented in PyMBAR in the case of the protein FEP calculations and by the standard error of the mean of replicas in the case of the membrane FEP calculations. All analysis scripts are available on GitHub[75].

We considered convergence of the calculations through several metrics. The cumulative change in free energy with respect to $\lambda$ are shown in Supplementary Figs. 17 and 21. We further tested convergence by comparing the cumulative $\Delta G$ values that resulted from using the first vs second half of the raw data, for each replica (Supplementary Figs. 18 and 22), and required that these values be within -1 kcal/mol of each other. Finally, internal hysteresis was assessed by calculating the difference $\delta_\lambda$ between the forward and backward exponential free energy estimates per window. Well-converged calculations are expected to have only small differences between forward and backward calculations (Supplementary Figs. 19 and 23) that are randomly distributed with respect to lambda (Supplementary Figs. 20 and 24). We used approximately minimal lambda schedules to ensure convergence thus defined; shorter, fewer windows were found to yield inadequate sampling of the ensembles.

## Pharmacological and allosteric predictions

The relative free energies of replacement calculated above (Eq. 4a, Table 1) can then be converted to relative binding constants:

$$K_M^{ab} = \frac{K_M^b}{K_M^a} = \exp\left(-\frac{\Delta\Delta G_{Bulk \to M}^{a \to b}}{RT}\right) \tag{4a}$$

Where $K_M^a$ is the absolute binding constant of lipid $a$ to state M, and $K_M^{ab}$ is the relative binding constant of lipid $b$ relative to that of lipid $a$. For example, the state-dependent binding constant for POPG binding to the ELIC5 conformation (relative to POPC) is:

$$K_{ELIC5}^{CG} = \frac{K_{ELIC5}^{PG}}{K_{ELIC5}^{PC}} = \exp\left(-\frac{\Delta\Delta G_{Bulk \to ELIC5}^{PC \to PG}}{RT}\right) \tag{4b}$$

The above can be extended to predictions of allosteric modulation if we make the following assumptions:

1. ELIC has at least two distinct conformations, which we denote ELIC5 and WT.
2. A maximum of three lipid species are present: POPE, POPC, and POPG. The sum of mole fractions of the three lipids is 1: $x_{POPC} + x_{POPE} + x_{POPG} = 1$
3. The allosteric effects of each binding site are independent of the other binding sites.
4. The probability of a site being unoccupied is negligible (i.e., when a lipid leaves the binding site it is rapidly replaced by another).

Under these assumptions, the probability $p_{occ,M}^a$ of a site in conformation M to be occupied by a lipid of species $a$ is

$$p_{occ,M}^a = \left(1 + K_M^{ab}\frac{x_b}{x_a} + K_M^{ac}\frac{x_c}{x_a}\right)^{-1}, \tag{5a}$$

For example, the fraction of sites occupied by POPC in the ELIC5 conformation is

$$p_{occ,ELIC5}^{PC} = \left(1 + K_{ELIC5}^{CG}\frac{x_{PG}}{x_{PC}} + K_{ELIC5}^{CE}\frac{x_{PE}}{x_{PG}}\right)^{-1}, \tag{5b}$$

The relative probability of observing the protein in each of the two conformations, as a function of lipid composition, is derived below. First, the law of mass action for a lipid binding to a given conformation is given by the product of the total amount of protein in that state, the equilibrium constant of binding, and the mole fraction of the lipid in the membrane. For example:

$$[ELIC5^{POPC}] = [ELIC5^\varnothing]K_{ELIC5}^{POPC}x_{PC}, \tag{6}$$

where $[ELIC5^{POPC}]$ and $[ELIC5^\varnothing]$ are the concentrations of ELIC in the ELIC5 state, with bound POPC and without a bound lipid, respectively. The probability ratio of observing ELIC in the ELIC5 conformation vs the WT conformation $\frac{p_{ELIC5}}{p_{WT}}$ is then:

$$\frac{p_{ELIC5}}{p_{WT}} = \frac{[ELIC5^\varnothing]}{[WT^\varnothing]} \cdot \frac{1 + K_{ELIC5}^{POPC}x_{PC} + K_{ELIC5}^{POPE}x_{PE} + K_{ELIC5}^{POPG}x_{PG}}{1 + K_{WT}^{POPC}x_{PC} + K_{WT}^{POPE}x_{PE} + K_{WT}^{POPG}x_{PG}}. \tag{7a}$$

We assume that the fraction of receptors with unoccupied sites is negligible, and switch to POPC-relative binding constants to obtain:

$$\frac{p_{ELIC5}}{p_{WT}} = \frac{p_{ELIC5}^{POPC}}{p_{WT}^{POPC}} \cdot \frac{x_{PC} + K_{ELIC5}^{CE}x_{PE} + K_{ELIC5}^{CG}x_{PG}}{x_{PC} + K_{WT}^{CE}x_{PE} + K_{WT}^{CG}x_{PG}}, \tag{7b}$$

This expression yields the ratio between the overall population of both conformations as the ratio obtained in pure POPC, multiplied by the modulation factor due to the presence of POPE and POPG lipids. We separate the latter contribution and take its logarithm, yielding the quantity plotted in Fig. 5c:

$$\Delta\log\left(\frac{p_{ELIC5}}{p_{WT}}\right) \equiv \log\frac{p_{ELIC5}}{p_{ELIC5}^{POPC}} - \log\frac{p_{WT}}{p_{WT}^{POPC}} = \log\frac{x_{PC} + K_{ELIC5}^{CE}x_{PE} + K_{ELIC5}^{CG}x_{PG}}{x_{PC} + K_{WT}^{CE}x_{PE} + K_{WT}^{CG}x_{PG}}. \tag{8}$$

## Statistical Information

Statistical analysis was performed as indicated in the figure legends. For the data in Fig. 1d, e and Supplementary Fig. 2, a one-way ANOVA with a post-hoc Tukey's test was performed. For the data in Figs. 4 and 6, the no exchange and exchange samples were compared using a paired, two-sided T-test. For the data in Supplementary Fig. 6c–e, WT and ELIC3 were compared using an unpaired, two-sided T-test. All data are reported as mean ± SEM. No sample size calculation as made.

## Reporting summary

Further information on research design is available in the Nature Portfolio Reporting Summary linked to this article.

# Data availability

The data supporting the findings of this study are available within the paper and supplementary information files and available from the corresponding author upon request. The cryo-EM maps have been deposited in the Electron Microscopy Data Bank (EMDB) under

accession codes EMD-27216 (WT CA POPC), EMD-27215 (WT Apo POPC), EMD-27218 (WT CA 2:1:1), EMD-27217 (WT Apo 2:1:1), EMD-27219 (ELIC3 CA 2:1:1), EMD-27220 (ELIC5 CA 2:1:1). The structural coordinates have been deposited in the RCSB Protein Data Bank (PDB) under the accession codes 8D64 (WT CA POPC), 8D63 (WT Apo POPC), 8D66 (WT CA 2:1:1), 8D65 (WT Apo 2:1:1), 8D67 (ELIC3 CA 2:1:1), 8D68 (ELIC5 CA 2:1:1). Initial model building of the ELIC CA 2:1:1 structure used the ELIC crystal structure, PDB 2YN6. Source data are provided with this paper.

## Code availability

FEP data and analysis scripts are available at https://github.com/ BranniganLab/ELIC-phospholipid-identification (https://doi.org/10. 5281/zenodo.7264978).

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

## Acknowledgements

This study was supported by grants R35GM137957 to W.C., F32GM139351 to J.P., NSF2152059 to G.B. and E.S.M., and K08GM139031 to T.J. We are grateful to Drs. Olaf Andersen and Philipp Schmidpeter for help in establishing the fluorescence stopped-flow flux assay and data analysis, and to Dr. Mark Arcario and Dr. Joe Henry Steinbach for useful discussions. We also acknowledge the Office of Advanced Research Computing (OARC) at Rutgers, The State University of New Jersey for providing access to the Caliburn cluster and associated research computing resources that have contributed to the results reported here. https://oarc.rutgers.edu.

## Author contributions

J.T.P. and W.W.C. conceived the project, designed the experimental procedures and drafted the paper. J.T.P., N.M.D., B.D., M.S.W., and L.J.C. carried out mutagenesis, protein purification and the stopped-flow flux measurements. K.T.M. and N.M.D. performed the giant liposome

patch-clamp recordings. J.T.P. performed the cryo-EM sample preparation and single particle analysis. M.R. and J.A.J.F. helped with grid screening and single particle analysis. P.Y. and Z.D. helped with structural model building and refinement. G.B., J.H., E.S.M., and T.T.J. performed the aFEP analysis. All authors reviewed the paper.

## Competing interests

The authors declare no competing interests.
