## [Peer Review File · Nature Communications]

Open-channel structure of a pentameric ligand-gated ion channel reveals a mechanism of leaflet-specific phospholipid modulationREVIEWER COMMENTS

Reviewer #1 (Remarks to the Author):

This study from Petroff et al. identifies a putative lipid modulatory site in a bacterial pentameric ligand-gated ion channel, ELIC, and tests its identity, interactions, and role in functional modulation of channel activity. This complementary approach yields convincing interpretations and was impressive to me. While headway on structural biology of the eukaryotic channels has resulted in somewhat diminished interest in the bacterial orthologs, here the authors are able to do what to date no one has been able to do, to my knowledge with the eukaryotic pLGICs, in terms of rigorously connecting density for a lipid to its identity and modulatory effect. Accordingly, I think this study is not only well executed but will also be appreciated broadly in the community.

Some comments/questions/suggestions:

1. "Open" is a word that describes a conformation while "Activated" denotes a functional state. The authors frequently talk about "the open state," when they intend to describe an activated state with an open channel. Please use this terminology consistently.
2. Another terminology request. Maps from EM are not electron density maps (as from x-ray crystallography).
3. The conformational transition for ELIC from a resting state to an activated state is particularly interesting, because it appears to be different from every other pLGIC I know of (ECD expands during activation for ELIC?). However, the figures illustrating this conformational change, using superpositions, are unclear. Fig 3c: are these global or single subunit superpositions, or superposition of one subunit to look at what the adjacent subunit is doing? Same questions for ED Fig 3 and 8. For ED Fig 3, also, I do not understand how pairs of structures are being compared. More is needed to clearly illustrate the proposed steps from resting to pre-active to activated. Complementing figures with morphing movies could help.
4. ED Fig 8c: it looks like the top of M4 is becoming disordered, or the whole helix translates, in a state-dependent manner. Please clarify.
5. Modeling (next few points). Thank you for sharing the maps and models. Overall the models look well built with excellent density features for the whole receptor either in sharpened or unsharpened maps depending on the region. One curiosity is why Gln233 is modeled as it is, specifically in the ELIC_5_mut_CA_5_POPG_5_CA model.
6. There is reasonably strong density (in sharpened map) for rotamer 1, and no density for it where it is modeled. Rotamer 1 would point it toward the 5-fold axis.

7. The lipid density is very noisy compared to the receptor, but the signal is convincing to me, and complemented by the functional assays, the fitting seems reasonable. Somewhat surprising is that in the unsharpened map, the lipid density is much worse than in the sharpened map; the opposite is usually true, with areas lacking high resolution features (like lipids and the M4 helix) getting better in low-pass filtered and/or unsharpened maps. Was a tight mask used to generate the unsharpened map, such that the lipid density was masked out?

8. There are a number of bond length/angle/torsion outliers in the pdb validation for the agonist and POPG. These may stem from improper values in the .cif ligand geometry file (has happened to us when using phenix elbow to generate restraints).

9. Description of lipid site. The authors describe the locus as within a groove formed by M4, M3 and M1. This made me initially think lipid is binding within a single subunit, between M4, M3 and M1. Instead, much of the lipid spans a subunit interface, interacting with M3 and M4 on the principal side, and M1 on the complementary side. Perhaps emphasize that it spans a subunit interface.

10. Page 9, zeta potentials. Coming from the structural biology side, I was not familiar with zeta potentials. Could you add a brief explanation of what these are and what they tell us in this application?

Reviewer #2 (Remarks to the Author):

- What are the noteworthy results?

The main results of the manuscript are that the first truly open state structure of the prokaryotic pentameric channel homologue ELIC is determined and in addition the mechanism for allosteric activation of ELIC by the anionic lipid POPG.

- Will the work be of significance to the field and related fields? How does it compare to the established literature? If the work is not original, please provide relevant references.

ELIC structures that have been published so far are of a non-conductive channel, whereas the engineering here yielded increasingly opening pores. In particular the wildtype ELIC structure without agonist is constricted at 16' and 9' with only some increase in the pore diameter at 16' in the presence of agonist. Here, however, the gain-of-function triple mutant in the presence of agonist is open much further at the 16' position and the entire channel (including the 9' position) is open in the ELIC construct with 5 positions engineered. This sequential opening may portray the actual sequence of opening in wildtype channels. The insights into the conformational wave from ligand binding at subunit interfaces in the extracellular domain with loop C clamping down and a concomitant translation away from the pore, beta-sheet expansion of beta1-beta6, and an outward blooming of TM helices that extent through the entirety of the TMD are described in detail and supported by tryptophane quenching as well.

In addition to the different pore conformations obtained by engineering ELIC, structural insights into the lipid binding site that was previously shown for phospholipids and cardiolipin (which is also negatively charged) indicate that conformational changes during channel opening make the lipid binding site more available and lead to higher occupancy as compared to apo or partially open structures. The lipid binding site is further explored with a set of single site substitutions for which the functional impact relates to their interaction with the lipid headgroup in the structures.

The outer leaflet lipid modulator site is also further investigated using asymmetric lipids.

Functionally, the ELIC5 construct reproduces the observed fully open structure, since the construct does not desensitize whereas wildtype desensitizes rapidly in the presence of agonist.

- Does the work support the conclusions and claims, or is additional evidence needed?

The manuscript in the present form represents an extensive and comprehensive amount of work. Stopped flow and flux assays in different lipid conditions, cryo-EM structures in different functional states, and MD simulations complement and corroborate each other.

- Is there enough detail provided in the methods for the work to be reproduced?

The methods are clearly and fully described to be reproduced.

Minor comment:

Replace Quickchange with Quikchange.

I am afraid I have no other suggestions for how to improve this manuscript.

Sincerely,

Michaela Jansen

Reviewer #3 (Remarks to the Author):

This is a strong manuscript focusing on the effect of PG lipids on the function of a model PGLIC (ELIC). The work combines a set of strong experimental measurements that is strongly supported by computational work that clearly shows the preferential binding of POPG to the protein in the outer

leaflet. Overall this work is worthy of publishing but details of equilibration of the protein structures should be provided.

General Concern:

Equilibration of systems: The MD methods state that at least 20ns of equilibration was performed before FEP simulations. There is no proof or detail of the range of equilibration time for the different systems. The authors should provide some details on how the protein structure changes during this short time to demonstrate that ~20ns is enough. If the authors are arguing that they want to maintain the structure of the specific state with this short equilibration, then this should be made clear in the manuscript.

Reviewer #4 (Remarks to the Author):

The manuscript by Petroff et al describes several structures of the model pentameric ligand-gated ion channel, ELIC, in different lipid environments along with considerable functional data aimed at elucidating the mechanisms by which lipids influence channel function. Of note, they report the first ELIC structures of both an open-state and a “partially open state”. Given that several agonist-bound structures of ELIC have been solved to date in closed conformations, the solution of an open state structure is a major achievement with important mechanistic consequences. This finding itself is worthy of publication. I strongly support the publication of this important manuscript but recommend that the manuscript be rewritten to focus on the key findings and soften more speculative interpretations.

Specific concerns:

1) The presented structures are the starting point for the entirety of this work. Given previous inability to stabilize an open state, these structures represent a major advance and are central to the findings – they need to take more of a central stage in the manuscript. A major weakness of the paper, as written, is the lack of attention given to the functional annotation of the structures, which diminishes the discussion of the structures in the context of ELIC/pLGIC mechanisms of activation.

Figure 2a shows that ELIC5 remains open indefinitely in the presence of agonist (i.e. under conditions similar to those that are used in the cryo-EM experiments) consistent with the assignment of ELIC5 CA

2:1:1 to the open state. Exposure of WT ELIC to agonist in 2:1:1 membranes, however, leads to rapid desensitization, yet the WT CA 2:1:1 structure recorded under these conditions is attributed to a pre-active state. ELIC3 also appears to remain open essentially indefinitely, but ELIC3 CA 2:1:1 is implied to be a different pre-active state – although this is not entirely clear. The annotation of the observed structures is not consistent with the available electrophysiology data - so there is a disconnect. The annotation of the structures needs to be explored, clarified, and critically rationalized. For the field of ELIC structural biology, these annotations must be accurate.

It would strengthen the manuscript to present a more detailed characterization of the electrophysiological properties (see below) of the WT and ELIC3 and ELIC5 mutants in the different lipid environments used to solve structures and then discuss the various structures in the context of the electrophysiological data. More effort needs to be devoted to interpreting the structures in the context of previously solved ELIC structures and those of other pLGICs. The authors need to discuss their structures in the context of the mechanisms of ELIC activation and desensitization. There is potentially a lot of useful information here that has not been explored or carefully rationalized.

2) The observation of an open state specific PG binding site is intriguing and worth pursuing, but this interpretation is complicated by many factors that are not discussed in the manuscript. First, the number and position of lipids bound to pLGICs in cryo-EM structures depends on many factors beyond conformational state, including the resolution of the structure, the nanodisc used to solubilize the protein, and the conditions for nanodisc preparation, and likely simple variations from one structure to another. (discussed in Ananchenko et al., 2022). This is particularly evident in the nine recent structures of the nAChR, with different structures showing different patterns of lipid binding even to the same conformational state. Second, Kumar et al (2021) see cardiolipin bound to a site that overlaps with open-state PG site identified here so there may be PG binding in the resting state, you may simply not see it in your structures. Third, the authors do not see PG binding to the intracellular leaflet site even though binding to that site has been characterized by both the author's lab and other labs. This reinforces the point that the absence of binding to a site in one structure does not necessarily mean that there is no lipid binding to that site – it just means that bound lipids to that site are not clearly seen in that particular structure. Third, several mutations were performed to stabilize the open state, including three aromatic insertions near the end of M4. MD simulations performed previously by one of the authors show that these three aromatic residues impact lipid binding in this region (see Carswell et al., Structure 2015). So it is not clear that the observed PG binding is a consequence of stabilization of the open state – it may be a consequence of the ELIC5 mutations. A more definitive interpretation would require a comparable structure of ELIC5 in the absence of agonist. Such a structure would also enhance the MD simulations reported later exploring the binding of PG to active versus inactive conformations. Finally, it is important to note that the authors show that WT ELIC desensitizes after agonist application even in the presence of PG, while ELIC3 and ELIC5 do not. So one cannot conclude that PG stabilizes an open state, it facilitates opening and then desensitization. In addition, previous studies have suggested that ELIC3 activates in POPC membranes lacking PG. A further experiment would thus be to record electrophysiological data for ELIC5 in POPC liposomes to see if PG is really required.

Although the observation of strong electron density in the open structure is intriguing and potentially important, it is very difficult to definitively state that there is an active state-specific PG binding site on ELIC and that this binding is critical for function – the discussion around this issue needs to be tempered given the above complications. Further work could be done to strengthen this conclusion.

3) One weakness of this paper is that the results are not discussed sufficiently in the context of previous work on ELIC-lipid interactions of Henault et al. (2019), Tong et al. (2019), and Kumar et al. (2021). In particular, Henault et al paper shows PG dramatically slows WT ELIC desensitization and presents strong evidence that PG binding to an intracellular leaflet site also slows desensitization. The possibility that the effects of PG are due to binding to this intracellular site needs to be taken into account in the discussion of the current thallium flux data (see below).

4) Although the authors are commended for the extensive thallium flux assays used to characterize ELIC function in different lipid environments, it would be helpful if the limitations of the techniques were acknowledged in the discussion. In all but one case, raw data demonstrating the effects of different lipids on ELIC activity is not presented – only processed data and conclusions. This reduces the impact of the work, particularly because it is not intuitive how the parameters measured in the bar graphs (time constant of decay and peak rates) relate to ion channel physiological properties, such as agonist affinity, gating equilibrium constants or channel open/closed times, channel conductance, rates of desensitization, etc. It is particularly difficult to interpret some of the processed data given the above noted fact that PG has a dramatic effect on ELIC desensitization kinetics (Henault et al., 2019). It would be helpful to discuss the flux data in the context of the known effects of PG on desensitization in the context of ion channel physiological properties.

Some of the conclusions from the flux studies are also overstated and confusing. For example, EC50 values are a composite of many kinetic parameters related to binding affinity and gating equilibrium constants, and measured values can be influenced by changes in the rates of desensitization. How can the authors conclude that (line 105) the lack of a change in EC50 between POPC or PC:PE:PG 2:1:1 means there is an absence of change in the equilibrium between “resting and agonist-bound states”, particularly from bulk measurements of flux into vesicles? The authors also conclude in the next line (line 108) that POPG modulates ELIC function by “stabilizing the open state of the channel relative to one or more agonist-bound non-conducting states”. If the open state is stabilized by POPG, then POPG must either change the energy (i.e. change the equilibrium) between agonist-bound open and resting states (in contrast to line 105) or if it does not change the equilibrium between resting and agonist-bound states (line 1050 then it must lower the energy of the resting state to the same extent as it lowers the energy of the open state – an assertion contradicting the main point of the paper that PG interacts preferentially with the open state.

Also, in Figure 2a, the authors show that 10 mM Cysteamine leads to rapid activation followed by (on the 10s of seconds time scale) rapid desensitization in the presence of PG. So it is incorrect to conclude

that POPG stabilizes the open state over agonist-bound closed states, such as the desensitized state, otherwise the WT CA 2:1:1 would be open. Based on the data, PG facilitates conformational transitions to the open and then desensitized states.

One final comment with respect flux studies. One of the early limitations of vesicle ion flux studies performed on the nAChR was that different lipid compositions lead to different vesicle sizes, which ultimately lead to variability in the measured flux response. While the authors have prepared giant unilamellar vesicles, it is not clear whether there are still differences in vesicle size with different lipid compositions and how these might impact on the findings. We have also found that changing membrane lipid composition can influence the % incorporation of the nAChR into a reconstituted membrane (daCosta & Baenziger, 2009). We eventually ended up purifying the proteoliposomes from non incorporated protein and empty vesicles using sucrose gradients. My experience is that the interpretation of vesicle flux measurements must be tempered in the context of their limitations.

I also wonder if treatment of vesicles with methyl- β -cyclodextrin could lead to changes in vesicle size and/or integrity. The authors state that they performed a control with empty methyl- β -cyclodextrin treatment, but I don't see the data. What about using other lipids, such as PC, as a further control in the methyl- β -cyclodextrin experiments? Is it possible that the initial rapid influx observed with the asymmetrically added PG is due to the methyl- β -cyclodextrin treatment damaging the vesicle integrity so that you get a rapid flux into damaged vesicles followed by the slower decay – which is similar to that observed in the non-PG vesicles?

Finally, Zeta potentials were used to assess whether or not the PG is incorporated into the outer leaflet, but is it possible that small amounts of PG flip to the other side to influence flux by the intracellular leaflet site? The latter possibility is hard to assess given that we have no sense as to how much PG must flip to the other side to have a phenotypic effect and whether the Zeta potentials can detect this amount. Although I am intrigued and tend to agree with the authors that extracellular leaflet PG is important, this interpretation is complicated given the potential limitations of the assays and the compelling data that PG binding to an intracellular leaflet site has a dramatic effect on ELIC desensitization. At least some discussion of this is required.

In summary, there is extensive, important and exciting data in this paper, but there are two distinct avenues that are explored as a result of the findings – structures pertaining to ELIC conformational transitions and data pertaining to lipid-ELIC interactions. I strongly support publication of this work but suggest that the authors need to first focus on interpreting properly the new structures in terms of what they tell us about the mechanisms of ELIC activation. This in itself represents a complete publication.

Minor concerns

1) The authors cite the recent boon of lipid-bound pLGICs structures (lines 69-71) specifically mentioning the 5-HT3AR, GABAAR and ELIC – yet ignore other structures, such as early studies of GLIC and the nine recent Torpedo structures, which reveal the most extensive details regarding lipid binding. Lipid binding to pLGICs has been reviewed by Thompson et al. (2020) and Ananchenko et al. (2022). It might be more appropriate to reference one of these reviews.

2) Line 64, please be clear what you mean by stability as this could be confused with thermal stability. Although you are referring to a conformational selection mechanism, other mechanisms by which lipids influence function are possible. For example, the nAChR adopts a lipid-dependent uncoupled state, which adds complexity to the interpretation of lipid-nAChR interactions.

3) There is not a consensus that PE is required for optimal activity of the nAChR (lines 89-90). The most compelling data is that anionic lipids and cholesterol are important.

4) Line 105, if it is not statistically significant, then it is within the error of the measurement – remove “2x higher”.

5) The authors conclude that PG stabilizes the open state because it facilitates activation of ELIC. The electrophysiological data showing that WT ELIC desensitizes rapidly refutes this. As noted above, PG facilitates ELIC transitioning to open and then desensitized states.

REVIEWER COMMENTS

Reviewer #1 (Remarks to the Author):

This study from Petroff et al. identifies a putative lipid modulatory site in a bacterial pentameric ligand-gated ion channel, ELIC, and tests its identity, interactions, and role in functional modulation of channel activity. This complementary approach yields convincing interpretations and was impressive to me. While headway on structural biology of the eukaryotic channels has resulted in somewhat diminished interest in the bacterial orthologs, here the authors are able to do what to date no one has been able to do, to my knowledge with the eukaryotic pLGICs, in terms of rigorously connecting density for a lipid to its identity and modulatory effect. Accordingly, I think this study is not only well executed but will also be appreciated broadly in the community.

Some comments/questions/suggestions:

1. “Open” is a word that describes a conformation while “Activated” denotes a functional state. The authors frequently talk about “the open state,” when they intend to describe an activated state with an open channel. Please use this terminology consistently.

We have modified the text throughout to have consistent and accurate terminology. With regards to channel states and structures, we now consistently use the term “activated, open-channel state”. With regards to structure and conformation, we now use the term “open-channel conformation” or “open-channel structure”.

2. Another terminology request. Maps from EM are not electron density maps (as from x-ray crystallography).

We have made this change.

3. The conformational transition for ELIC from a resting state to an activated state is particularly interesting, because it appears to be different from every other pLGIC I know of (ECD expands during activation for ELIC?). However, the figures illustrating this conformational change, using superpositions, are unclear. Fig 3c: are these global or single subunit superpositions, or superposition of one subunit to look at what the adjacent subunit is doing? Same questions for ED Fig 3 and 8. For ED Fig 3, also, I do not understand how pairs of structures are being compared. More is needed to clearly illustrate the proposed steps from resting to pre-active to activated. Complementing figures with morphing movies could help.

The reviewer is correct in recognizing that the ECD expansion observed in the ELIC open structure is unique to ELIC when compared with available pLGIC structures. Fig. 3, ED Fig. 3 (now Supplementary Fig. 5) and ED Fig. 8 (now Supplementary Fig. 11) show global superpositions of the structures in all cases. In ED Fig. 3 (now Supplementary Fig. 5), we are comparing different pairs of structures within each subunit to efficiently convey differences between the many structures presented in this study. This was generated by creating a global superposition of all structures and then displaying different pairs for each of the pentameric subunits. We have clarified this in the figure legends for Fig. 3 and now Supplementary Fig. 5 and 11. Also, we have generated morphing movies of the transition from WT apo to WT CA and WT CA to ELIC5 CA (Supplementary Movies 1-8). This illustrates the conformational transitions between these key structures in both the ECD, ECD-TMD interface and TMD.

4. ED Fig 8c: it looks like the top of M4 is becoming disordered, or the whole helix translates, in a state-dependent manner. Please clarify.

This is indeed the case. We have added content in the results to state that the cryo-EM density of M4, especially at the top, is absent in all structures except for ELIC5 CA. Therefore, we modeled only up to residue 317 in all structures except for ELIC5 CA, which was modeled to residue 321. This suggests that M4 is more dynamic in resting and agonist-bound non-conducting states than in the open state. We also expanded the description of structural changes of M4 in ELIC5 CA (lines 238-246). The helix translates outward 4.7 Å from WT apo and WT CA to ELIC5 CA, and straightens below P305.

5. Modeling (next few points). Thank you for sharing the maps and models. Overall the models look well built with excellent density features for the whole receptor either in sharpened or unsharpened maps depending on the region. One curiosity is why Gln233 is modeled as it is, specifically in the ELIC_5_mut_CA_5_POPG_5_CA model.

See response below.

6. There is reasonably strong density (in sharpened map) for rotamer 1, and no density for it where it is modeled. Rotamer 1 would point it toward the 5-fold axis.

The reviewer is correct in identifying a strong density for rotamer 1 of Q233 in the ELIC5 CA structure. We modeled Q233 as is in the structure because there is also density, albeit weaker, that appears to provide a good fit for the alternative rotamer. To address this, we added rotamer 1 to the model. Since the density for rotamer 1 appears to be stronger, we assigned an occupancy of 70% for rotamer 1 and 30% for rotamer 2. We added Supplementary Fig. 8 to show the cryo-EM density for both rotamers and the pore profile of the structure that uses rotamer 1 for Q233. Also, we state (lines 161-170) that the two conformations yield minimum pore diameters of 7.3 Å and 5.6 Å at 2'. The larger diameter is consistent with the reported low conductance of quaternary ammonium cations in ELIC (especially tetraethylammonium). More work is needed to characterize the ELIC5 open structure and the potential role of Q233 in determining ELIC ion conduction properties. Here we referenced three studies from Claudio Grosman's group demonstrating the importance of side chain rotamers in this region of the pore in other pLGICs.

7. The lipid density is very noisy compared to the receptor, but the signal is convincing to me, and complemented by the functional assays, the fitting seems reasonable. Somewhat surprising is that in the unsharpened map, the lipid density is much worse than in the sharpened map; the opposite is usually true, with areas lacking high resolution features (like lipids and the M4 helix) getting better in low-pass filtered and/or unsharpened maps. Was a tight mask used to generate the unsharpened map, such that the lipid density was masked out?

This is an interesting observation. The unsharpened map was not generated with a tight mask that would remove the lipid density. It is not clear to us why the lipid density is worse in the unsharpened map.

8. There are a number of bond length/angle/torsion outliers in the pdb validation for the agonist and POPG. These may stem from improper values in the .cif ligand geometry file (has happened to us when using phenix elbow to generate restraints).

We are grateful to the reviewer for providing additional instruction for correcting these outliers. We have produced models of the DHL (cysteamine) ligand as instructed in all agonist-bound structures and the outliers were corrected in the validation report. However, we followed the same procedure for PGW in the ELIC5 structure and found that the outliers, especially torsion outliers persisted. Therefore, we have chosen not to modify the lipid in the lipid-bound structures. Of note, the bond length and angle outliers are relatively small. From a computational standpoint, these outliers will be quickly resolved during the initial energy minimization for the MD simulations. Therefore, we do not think the outliers impact the use of these models for future simulation studies, nor do they affect the interpretation/conclusions of the structural data.

9. Description of lipid site. The authors describe the locus as within a groove formed by M4, M3 and M1. This made me initially think lipid is binding within a single subunit, between M4, M3 and M1. Instead, much of the lipid spans a subunit interface, interacting with M3 and M4 on the principal side, and M1 on the complementary side. Perhaps emphasize that it spans a subunit interface.

We have modified the text to describe this site as an intersubunit site (lines 85, 185, 197, 335, and 343).

10. Page 9, zeta potentials. Coming from the structural biology side, I was not familiar with zeta potentials. Could you add a brief explanation of what these are and what they tell us in this application?

We have added a brief explanation of zeta potentials and its utility in this application in the Methods section (lines 592-596). Interested readers can refer to the Langmuir paper.

Reviewer #2 (Remarks to the Author):

• What are the noteworthy results?

The main results of the manuscript are that the first truly open state structure of the prokaryotic pentameric channel homologue ELIC is determined and in addition the mechanism for allosteric activation of ELIC by the anionic lipid POPG.

• Will the work be of significance to the field and related fields? How does it compare to the established literature? If the work is not original, please provide relevant references. ELIC structures that have been published so far are of a non-conductive channel, whereas the engineering here yielded increasingly opening pores. In particular the wildtype ELIC structure without agonist is constricted at 16' and 9' with only some increase in the pore diameter at 16' in the presence of agonist. Here, however, the gain-of-function triple mutant in the presence of agonist is open much further at the 16' position and the entire channel (including the 9' position) is open in the ELIC construct with 5 positions engineered. This sequential opening may portray the actual sequence of opening in wildtype channels. The insights into the conformational wave from ligand binding at subunit interfaces in the extracellular domain with loop C clamping down and a concomitant translation away from the pore, beta-sheet expansion of beta1-beta6, and an outward blooming of TM helices that extent through the entirety of the TMD are described in detail and supported by tryptophane quenching as well.

In addition to the different pore conformations obtained by engineering ELIC, structural insights into the lipid binding site that was previously shown for phospholipids and cardiolipin (which is

also negatively charged) indicate that conformational changes during channel opening make the lipid binding site more available and lead to higher occupancy as compared to apo or partially open structures. The lipid binding site is further explored with a set of single site substitutions for which the functional impact relates to their interaction with the lipid headgroup in the structures. The outer leaflet lipid modulator site is also further investigated using asymmetric lipids. Functionally, the ELIC5 construct reproduces the observed fully open structure, since the construct does not desensitize whereas wildtype desensitizes rapidly in the presence of agonist.

- Does the work support the conclusions and claims, or is additional evidence needed?
The manuscript in the present form represents an extensive and comprehensive amount of work. Stopped flow and flux assays in different lipid conditions, cryo-EM structures in different functional states, and MD simulations complement and corroborate each other.
- Is there enough detail provided in the methods for the work to be reproduced?
The methods are clearly and fully described to be reproduced.

Minor comment:

Replace Quickchange with Quikchange.

This has been changed.

I am afraid I have no other suggestions for how to improve this manuscript.

Sincerely,
Michaela Jansen

Reviewer #3 (Remarks to the Author):

This is a strong manuscript focusing on the effect of PG lipids on the function of a model PGLIC (ELIC). The work combines a set of strong experimental measurements that is strongly supported by computational work that clearly shows the preferential binding of POPG to the protein in the outer leaflet. Overall this work is worthy of publishing but details of equilibration of the protein structures should be provided.

General Concern:

Equilibration of systems: The MD methods state that at least 20ns of equilibration was performed before FEP simulations. There is no proof or detail of the range of equilibration time for the different systems. The authors should provide some details on how the protein structure changes during this short time to demonstrate that ~20ns is enough. If the authors are arguing that they want to maintain the structure of the specific state with this short equilibration, then this should be made clear in the manuscript.

We apologize for the lack of clarity on the purpose of the 20 ns MD simulations. The reviewer is correct that we intentionally do not want either the lipid or the protein to drift significantly away from the starting structure, because we are aiming to identify the most likely bound lipid that corresponds to a particular experimentally-determined density. We have clarified the purpose of these simulations in the caption to Figure 5 as well as in Methods (lines 618-624), which now read as follows:

Following the default relaxation scheme of CHARMM-GUI[62,66], systems were further relaxed for at least 20 more nanoseconds to ensure equilibration and to determine the expected deviation of the glycerol backbone due to thermal fluctuations around the modeled binding pose. As described below, these simulations are essential for parameterizing the Distance-to-Bound-Configuration (DBC) coordinate used in the SAFEP approach, but are not intended to thoroughly sample the full range of configurations available to the lipid or protein.

Reviewer #4 (Remarks to the Author):

The manuscript by Petroff et al describes several structures of the model pentameric ligand-gated ion channel, ELIC, in different lipid environments along with considerable functional data aimed at elucidating the mechanisms by which lipids influence channel function. Of note, they report the first ELIC structures of both an open-state and a “partially open state”. Given that several agonist-bound structures of ELIC have been solved to date in closed conformations, the solution of an open state structure is a major achievement with important mechanistic consequences. This finding itself is worthy of publication. I strongly support the publication of this important manuscript but recommend that the manuscript be rewritten to focus on the key findings and soften more speculative interpretations.

Specific concerns:

1) The presented structures are the starting point for the entirety of this work. Given previous inability to stabilize an open state, these structures represent a major advance and are central to the findings – they need to take more of a central stage in the manuscript. A major weakness of the paper, as written, is the lack of attention given to the functional annotation of the structures, which diminishes the discussion of the structures in the context of ELIC/pLGIC mechanisms of activation.

We have chosen to craft this manuscript with an emphasis on both a putative mechanism of phospholipid modulation and the structural insights gained from the open structure of ELIC. The features associated with the open structure are comprehensively detailed in the manuscript, with much of this detail presented in the context of the bound phospholipid. However, we agree with the reviewer that the open structure should be further emphasized. To highlight the finding of the open structure, we altered the title to include mention of the open structure. It now reads: “Open structure of a pentameric ligand-gated ion channel reveals a mechanism of leaflet-specific phospholipid modulation”. In addition, we have: 1) added discussion on the functional annotation of the structures, and 2) added movies (Supplementary Movies 1-8) and text to enhance presentation of the conformational changes associated with the open structure. Please see below for more details.

Figure 2a shows that ELIC5 remains open indefinitely in the presence of agonist (i.e. under conditions similar to those that are used in the cryo-EM experiments) consistent with the assignment of ELIC5 CA 2:1:1 to the open state. Exposure of WT ELIC to agonist in 2:1:1 membranes, however, leads to rapid desensitization, yet the WT CA 2:1:1 structure recorded under these conditions is attributed to a pre-active state. ELIC3 also appears to remain open essentially indefinitely, but ELIC3 CA 2:1:1 is implied to be a different pre-active state – although this is not entirely clear. The annotation of the observed structures is not consistent

with the available electrophysiology data - so there is a disconnect. The annotation of the structures needs to be explored, clarified, and critically rationalized. For the field of ELIC structural biology, these annotations must be accurate.

We recognize that there is a potential disconnect between the functional data (both patch-clamp and stopped-flow TI^+ flux) for WT and ELIC3, which show robust agonist responses followed by desensitization in 2:1:1 membranes (note that ELIC3 shows slower and less desensitization than WT but still desensitizes profoundly- Supplementary Fig. 6c and 6d), and the agonist-bound structures of WT and ELIC3. While we expect to observe desensitized conformations for these structures, the structures instead appear to be most consistent with pre-active states. This is based on: 1) structures of pLGICs, especially of the nAChR, showing that 16' and usually 9' are activation gates, 2) structures of pLGICs showing that desensitization is associated with a constriction at the intracellular pore, and 3) NMR studies of ELIC suggesting a constriction at the bottom of the pore in the desensitized conformation (Kinde et al. 2015. *Structure*. 23(6): 995-1004.). We speculate that an un-identified factor influencing the cryo-EM structures is producing this discrepancy, although it is also possible that these structures represent desensitized states in ELIC. Therefore, we have adjusted the text to simply conclude that the WT CA and ELIC3 CA structures are agonist-bound non-conducting structures. We recognize that these structures cannot be confidently annotated with the available data, and state that more work is necessary to clarify the annotation of these structures. We have modified the Results accordingly in lines 171-183. Subsequently, discussion of lipid modulation is made stating that WT CA is an agonist-bound non-conducting conformation.

It would strengthen the manuscript to present a more detailed characterization of the electrophysiological properties (see below) of the WT and ELIC3 and ELIC5 mutants in the different lipid environments used to solve structures and then discuss the various structures in the context of the electrophysiological data. More effort needs to be devoted to interpreting the structures in the context of previously solved ELIC structures and those of other pLGICs. The authors need to discuss their structures in the context of the mechanisms of ELIC activation and desensitization. There is potentially a lot of useful information here that has not been explored or carefully rationalized.

As mentioned above, we have discussed the WT, ELIC3 and ELIC5 structures considering the available functional data. Within the scope of this study, we do not think that additional functional experiments are necessary (see below). In addition, we wish to highlight pre-existing and newly added discussion of the structures in the context of solved ELIC structures and other pLGIC structures. These include statements that: 1) the WT structures are indistinguishable from previously determined ELIC structures in POPC nanodiscs (lines 127-132), 2) the conformational changes in the ECD-TMD interface and TMD associated with channel activation and opening in ELIC are similar to other pLGICs, including the nAChR (lines 199-207, 231-246, 396-408), and 3) the outward expansion of the ECD in the ELIC5 open structure is distinct from what has been reported in other pLGICs (lines 400-404). We have also noted that the outward blooming of the transmembrane helices in ELIC5 is more profound than what was reported in the open alpha7 nAChR structure (lines 401-404), and that M4 shows interesting changes in the conformation and dynamics between the non-conducting and ELIC5 structures such as a straightening of the helix below P305 (lines 238-246). This conformation of M4 in the open structure is distinct from what was reported in Henault et al 2019 for a desensitized conformation. To enhance understanding of these conformational changes associated with channel activation, we have added movies demonstrating the transition from non-conducting (WT apo and WT CA) to open (ELIC5 CA) conformations. To summarize, our discussion of the structures focusses on the insights gained from the ELIC5 open structure. These reveal

conserved conformational changes in the agonist-binding site, ECD-TMD interface, and TMD associated with channel activation, which ultimately lead to pore opening due to a global outward expansion of the TMD helices. On the other hand, the ELIC5 open structure shows an outward expansion of the ECD that appears to be a departure from other activated pLGIC structures, to date.

2) The observation of an open state specific PG binding site is intriguing and worth pursuing, but this interpretation is complicated by many factors that are not discussed in the manuscript. First, the number and position of lipids bound to pLGICs in cryo-EM structures depends on many factors beyond conformational state, including the resolution of the structure, the nanodisc used to solubilize the protein, and the conditions for nanodisc preparation, and likely simple variations from one structure to another. (discussed in Ananchenko et al., 2022). This is particularly evident in the nine recent structures of the nAChR, with different structures showing different patterns of lipid binding even to the same conformational state. Second, Kumar et al (2021) see cardiolipin bound to a site that overlaps with open-state PG site identified here so there may be PG binding in the resting state, you may simply not see it in your structures. Third, the authors do not see PG binding to the intracellular leaflet site even though binding to that site has been characterized by both the author's lab and other labs. This reinforces the point that the absence of binding to a site in one structure does not necessarily mean that there is no lipid binding to that site – it just means that bound lipids to that site are not clearly seen in that particular structure. Third, several mutations were performed to stabilize the open state, including three aromatic insertions near the end of M4. MD simulations performed previously by one of the authors show that these three aromatic residues impact lipid binding in this region (see Carswell et al., Structure 2015). So it is not clear that the observed PG binding is a consequence of stabilization of the open state – it may be a consequence of the ELIC5 mutations. A more definitive interpretation would require a comparable structure of ELIC5 in the absence of agonist. Such a structure would also enhance the MD simulations reported later exploring the binding of PG to active versus inactive conformations. Finally, it is important to note that the authors show that WT ELIC desensitizes after agonist application even in the presence of PG, while ELIC3 and ELIC5 do not. So, one cannot conclude that PG stabilizes an open state, it facilitates opening and then desensitization. In addition, previous studies have suggested that ELIC3 activates in POPC membranes lacking PG. A further experiment would thus be to record electrophysiological data for ELIC5 in POPC liposomes to see if PG is really required.

Although the observation of strong electron density in the open structure is intriguing and potentially important, it is very difficult to definitively state that there is an active state-specific PG binding site on ELIC and that this binding is critical for function – the discussion around this issue needs to be tempered given the above complications. Further work could be done to strengthen this conclusion.

The reviewer raises important points regarding caveats to our interpretation of a lipid density in the ELIC5 CA structure. We acknowledge there are limitations to the interpretation of this structural data alone, and have addressed the reviewer's points in the next paragraph. In addition, it is important to note that this structural finding was then the rationale for performing an extensive alchemical free energy perturbation analysis to test selectivity of lipid binding to this site and state-dependence of binding. In concert with our asymmetric liposome and mutagenesis data, we believe the data support POPG modulation through this site, while not excluding the possibility of modulation also through another site (lines 270-272). We have tempered our conclusions by stating that these findings “suggest” or “support” a mechanism of modulation through this site (lines 194, 343, and 346), while stating the caveats described below.

The following is a specific response to each of the reviewer's concerns.

1) We agree that the presence of a lipid density depends on multiple factors. Therefore, we state that the lipid density suggests higher occupancy of a lipid in that conformation/state, and we performed aFEP analysis to test lipid specificity and state-dependence. We have added a statement that the presence of a lipid density at a site in one conformation does not exclude lipid interactions at this site in other conformations or at other sites (lines 344-347), and have referenced the muscle nAChR study which shows lipids occupying many different sites in different structures (lines 341).

2) We discussed the CL-bound ELIC structure noting that this is an apo structure of ELIC (lines 350-358). CL occupancy of this site in a resting conformation does not exclude the possibility that CL could also interact with the open conformation with a different binding mode and with a different probability of occupancy.

3) We have referenced studies from our laboratory (Tong et al 2019) and others (Henault et al. 2019) that PG binds to and could modulate ELIC through an inner leaflet site (line 272). This was noted when describing the asymmetric liposome data, recognizing that our findings do not exclude the possibility that PG could also modulate ELIC from the inner leaflet. Indeed, we do not claim that PG does not interact at other sites; only that the site described in this study appears to be selective for PG and show state-dependent PG binding.

3- #2) We acknowledge that the changes in lipid/POPG binding observed in the structure and aFEP calculations could be a result of the ELIC5 mutations, although the mutated residues are too far from the phospholipid to make direct interactions. We conclude that the changes in lipid binding at this site are likely a consequence of stabilization of an apparent open conformation, recognizing the caveat about the mutations. We have stated this caveat in lines 347-350.

Final point) Although WT and ELIC3 desensitize profoundly, we find using the stopped-flow assay (and previously by giant liposome patch-clamp recordings) that POPG stabilizes the open state of the channel relative to the desensitized state because POPG decreases the extent of desensitization. While most of the channels are still desensitized at steady state, POPG appears to shift the relative stability of these states in favor of the open state. We have clarified this interpretation in lines 105-108 and 112-113, referencing previous studies of Tong et al. eLife and Henault et al. NCB. In addition, we acknowledge that POPG facilitates ELIC activation and have modified our statements of POPG effects on ELIC activation accordingly (lines 108-112, 113-115). While testing the lipid sensitivity of ELIC5 (POPC versus 2:1:1 membranes) could be an interesting experiment, we do not think performing this experiment is critical to accept the findings described above.

3) One weakness of this paper is that the results are not discussed sufficiently in the context of previous work on ELIC-lipid interactions of Henault et al. (2019), Tong et al. (2019), and Kumar et al. (2021). In particular, Henault et al paper shows PG dramatically slows WT ELIC desensitization and presents strong evidence that PG binding to an intracellular leaflet site also slows desensitization. The possibility that the effects of PG are due to binding to this intracellular site needs to be taken into account in the discussion of the current thallium flux data (see below).

As noted above, we have referenced at multiple points the findings of previous work on ELIC-lipid modulation and interactions, as it relates to the current study. In particular, we state how

our stopped-flow findings are consistent with previous studies (Henault et al and Tong et al) showing that PG reduces the rate and extent of desensitization; studies which present evidence for POPG modulation through an inner leaflet site. While we believe that the asymmetric liposome assay results strongly argue for an outer leaflet mechanism, we specifically state that an inner leaflet mechanism is also possible, referencing Henault et al and Tong et al (lines 270-272).

4) Although the authors are commended for the extensive thallium flux assays used to characterize ELIC function in different lipid environments, it would be helpful if the limitations of the techniques were acknowledged in the discussion. In all but one case, raw data demonstrating the effects of different lipids on ELIC activity is not presented – only processed data and conclusions. This reduces the impact of the work, particularly because it is not intuitive how the parameters measured in the bar graphs (time constant of decay and peak rates) relate to ion channel physiological properties, such as agonist affinity, gating equilibrium constants or channel open/closed times, channel conductance, rates of desensitization, etc. It is particularly difficult to interpret some of the processed data given the above noted fact that PG has a dramatic effect on ELIC desensitization kinetics (Henault et al., 2019). It would be helpful to discuss the flux data in the context of the known effects of PG on desensitization in the context of ion channel physiological properties.

The presentation of the TI+ flux data conforms to the standard of the field using this approach in ligand-gated ion channels (ELIC, GLIC, CNG channels, KcsA). In many cases (WT and mutants), we present the full time course of channel activities (i.e. rates) in response to agonist at delay times ranging from 10 ms to 25 s. This captures ELIC activation and desensitization. The data are then processed to report the peak response and the time constant of desensitization. To further improve presentation of the data, we have added panel “a” to Figure 1 to illustrate the stopped-flow configuration and to show representative raw fluorescence quenching traces. We also modified Fig. 4a to show more raw traces. Furthermore, we have added details in the Methods regarding analysis of the fluorescence quenching data so that it is clear to the reader that we are measuring quenching rates (lines 541-555). Lastly, we added an inset to Fig. 1b that zooms into the activation time course. These show that activation in general is quite fast (time constants of tens of milliseconds) relative to the rates of desensitization (time constants of seconds). We agree that instead of using the term “decay in channel activity” we should state desensitization and have changed the text and figures accordingly. POPG slows the rate of desensitization (Fig. 1), consistent with previous findings regarding the effect of PG on desensitization kinetics (Henault et al, 2019 and Tong et al, 2019). The effect of POPG on peak response indicates an increase in peak channel activity (which is most likely related to open probability). However, we recognize that the available data does not provide detailed information regarding agonist affinity and gating equilibrium constants. We conclude, as recommended by the reviewer, that the functional data indicate that POPG stabilizes the open state of ELIC relative to a desensitized state, and that PG and PE facilitate channel activation leading to an increased peak response. This latter effect of PG and PE may reflect an increase in gating efficacy, although as noted by the reviewer, we cannot be certain since PG also affects channel desensitization (Note however that activation is overall much faster than desensitization). Based on the data, we hold that POPG stabilizes the open state of the channel relative to desensitized states. The effect of POPG on peak response suggests an increase in gating efficacy although it may also be a consequence of a change in desensitization. The discussion of the functional results have been modified accordingly in lines 105-120.

Some of the conclusions from the flux studies are also overstated and confusing. For example, EC50 values are a composite of many kinetic parameters related to binding affinity and gating

equilibrium constants, and measured values can be influenced by changes in the rates of desensitization. How can the authors conclude that (line 105) the lack of a change in EC50 between POPC or PC:PE:PG 2:1:1 means there is an absence of change in the equilibrium between “resting and agonist-bound states”, particularly from bulk measurements of flux into vesicles? The authors also conclude in the next line (line 108) that POPG modulates ELIC function by “stabilizing the open state of the channel relative to one or more agonist-bound non-conducting states”. If the open state is stabilized by POPG, then POPG must either change the energy (i.e. change the equilibrium) between agonist-bound open and resting states (in contrast to line 105) or if it does not change the equilibrium between resting and agonist-bound states (line 105) then it must lower the energy of the resting state to the same extent as it lowers the energy of the open state – an assertion contradicting the main point of the paper that PG interacts preferentially with the open state.

We agree with the reviewer that these statements need to be amended. First, we wish to clarify that the EC50 values are taken from peak response values obtained from full time courses obtained at each propylamine concentration. To illustrate this better, we have included the full time courses for each concentration in Supplementary Fig. 1. Indeed, the EC50 values derived from the peak response are a composite of many kinetic parameters. While there is a suggestion of a right-shift in the EC50 in POPC liposomes compared to 2:1:1, this is not statistically significant (lines 110-112). We have removed the statement that the dose response data informs the relative stability of resting and activated states between POPC and 2:1:1 membranes.

Also, in Figure 2a, the authors show that 10 mM Cysteamine leads to rapid activation followed by (on the 10s of seconds time scale) rapid desensitization in the presence of PG. So it is incorrect to conclude that POPG stabilizes the open state over agonist-bound closed states, such as the desensitized state, otherwise the WT CA 2:1:1 would be open. Based on the data, PG facilitates conformational transitions to the open and then desensitized states.

This comment has been addressed above. We believe that our stopped-flow data in conjunction from previous giant liposome patch-clamp measurements from Henault et al and Tong et al are a convincing body of evidence that POPG stabilizes the open state relative to the desensitized state. We agree that PG facilitates channel opening.

One final comment with respect flux studies. One of the early limitations of vesicle ion flux studies performed on the nAChR was that different lipid compositions lead to different vesicle sizes, which ultimately lead to variability in the measured flux response. While the authors have prepared giant unilamellar vesicles, it is not clear whether there are still differences in vesicle size with different lipid compositions and how these might impact on the findings. We have also found that changing membrane lipid composition can influence the % incorporation of the nAChR into a reconstituted membrane (daCosta & Baenziger, 2009). We eventually ended up purifying the proteoliposomes from non incorporated protein and empty vesicles using sucrose gradients. My experience is that the interpretation of vesicle flux measurements must be tempered in the context of their limitations.

The reviewer brings up good points, which we have addressed in the manuscript. Regarding liposome size, we have acquired cryo-EM images of POPC and 2:1:1 liposomes prepared for the stopped-flow assay to assess the size distribution of the liposomes. Measurements of liposome size from these images (representative images shown in Supplementary Fig. 3) show that there is some variation in liposome size but that the mean (~70 nm) is essentially the same in both lipid conditions. We conclude that it is unlikely that liposome size accounts for the

differences we see in agonist responses (lines 118-120). The reviewer is correct in stating that reconstitution efficiency is important; in addition, relative channel orientation is also important in determining peak response, since only outwardly facing channels will be activated. Importantly, the asymmetric liposome experiment controls for these factors because the “exchanged” and “no-exchanged” samples are taken from the same reconstitution. We have stated that based on the asymmetric liposome experiments, the effect of PG on peak response is likely a consequence of channel activity and not a change in the number of outwardly facing channels (lines 115-118, 265-269). Moreover, in a follow-up unpublished study, we have employed a fluorescent labeling method of the extracellular domain to independently measure the quantity of outwardly facing channels in every proteoliposome preparation. The results of these experiments are a significant body of work using a different ELIC construct that are beyond the scope of the current manuscript; nevertheless, they further confirm that the effects of PG are not a result of differences in reconstitution efficiency and channel orientation.

I also wonder if treatment of vesicles with methyl- β -cyclodextrin could lead to changes in vesicle size and/or integrity. The authors state that they performed a control with empty methyl- β -cyclodextrin treatment, but I don't see the data. What about using other lipids, such as PC, as a further control in the methyl- β -cyclodextrin experiments? Is it possible that the initial rapid influx observed with the asymmetrically added PG is due to the methyl- β -cyclodextrin treatment damaging the vesicle integrity so that you get a rapid flux into damaged vesicles followed by the slower decay – which is similar to that observed in the non-PG vesicles?

In the asymmetric liposome experiment, we added the same concentration of methyl- β -cyclodextrin to both the “exchange” and “no exchange”. This means that the effects on channel activity observed in the “exchange” sample are not simply a result of methyl- β -cyclodextrin compromising vesicle integrity. In fact, we find that in all samples treated with methyl- β -cyclodextrin, the retention of intraliposomal fluorescent Ti^+ indicator, ANTS, is unaffected, which also shows that vesicle integrity is not compromised (lines 379-381). We have modified the schematic in Fig. 4a to clarify that methyl- β -cyclodextrin was also added to the “no exchange” sample. Lastly, we added Supplementary Fig. 12 showing a comparison of ELIC responses in 3:1 POPC:POPE liposomes with and without methyl- β -cyclodextrin (lines 262-263). There is no significant difference in ELIC responses with the addition of methyl- β -cyclodextrin.

Finally, Zeta potentials were used to assess whether or not the PG is incorporated into the outer leaflet, but is it possible that small amounts of PG flip to the other side to influence flux by the intracellular leaflet site? The latter possibility is hard to assess given that we have no sense as to how much PG must flip to the other side to have a phenotypic effect and whether the Zeta potentials can detect this amount. Although I am intrigued and tend to agree with the authors that extracellular leaflet PG is important, this interpretation is complicated given the potential limitations of the assays and the compelling data that PG binding to an intracellular leaflet site has a dramatic effect on ELIC desensitization. At least some discussion of this is required.

The zeta potentials report on the amount of outer leaflet POPG, and therefore how much could be in the inner leaflet based on how much POPG was added to the sample (see Markones et al. 2018. *Langmuir*. 34:1999-2005.). The zeta potentials of the asymmetric liposome samples (“exchange” sample) never differed significantly from the symmetric liposome control (2:1:1) (Fig. 4b, Supplementary Fig. 15). Although there is some variability in these measurements, differences of no more than 3 mV were noted between asymmetric samples and control. Based on calibration curves generated from Markones et al. 2018 and Markones et al. 2020. *Biophysical Journal*. 118:294-302, we estimate that inner leaflet POPG is likely negligible and

no more than ~3-4 mole%. In contrast, we find that the ~25% outer leaflet POPG in the “exchange” sample mimics the effect of 25% POPG in symmetric liposomes, especially on the rate of desensitization. For example, in Fig. 1d and 1e, we show that 25% POPG in symmetric liposomes produces a time constant of desensitization of ~6s (compared to ~2.5s in 3:1 PC:PE) while 50% POPG produces a time constant of desensitization of ~12s. Similarly, in the asymmetric liposome experiments, introducing 25% POPG to the outer leaflet increases the time constant of desensitization from ~3 to ~6s. Therefore, the data strongly suggests that the observed modulatory effect of POPG especially on desensitization is produced by outer leaflet POPG, and not by any small amount of inner leaflet POPG. We have added this discussion to lines 372-379.

In summary, there is extensive, important and exciting data in this paper, but there are two distinct avenues that are explored as a result of the findings – structures pertaining to ELIC conformational transitions and data pertaining to lipid-ELIC interactions. I strongly support publication of this work but suggest that the authors need to first focus on interpreting properly the new structures in terms of what they tell us about the mechanisms of ELIC activation. This in itself represents a complete publication.

Minor concerns

1) The authors site the recent boon of lipid-bound pLGICs structures (lines 69-71) specifically mentioning the 5-HT₃AR, GABA_AAR and ELIC – yet ignore other structures, such as early studies of GLIC and the nine recent Torpedo structures, which reveal the most extensive details regarding lipid binding. Lipid binding to pLGICs has been reviewed by Thompson et al. (2020) and Ananchenko et al. (2022). It might be more appropriate to reference one of these reviews.

References for the Torpedo structure and the pLGIC-lipid reviews have been added to line 71, 73, 341, 385, and 404.

2) Line 64, please be clear what you mean by stability as this could be confused with thermal stability. Although you are referring to a conformational selection mechanism, other mechanisms by which lipids influence function are possible. For example, the nAChR adopts a lipid-dependent uncoupled state, which adds complexity to the interpretation of lipid-nAChR interactions.

We changed this to “conformational stability” to avoid confusion with thermal stability. We acknowledge that a possible mechanism by which POPG and POPE facilitate channel activation in ELIC is by promoting a coupled conformation. However, the current structures of ELIC in POPC and 2:1:1 membranes do not indicate the presence of coupled and uncoupled conformations. Also, FTIR measurements of ELIC in POPC membranes indicate that ELIC does not adopt the same uncoupled conformation as has been observed in the nAChR (Carswell et al. JBC 2015). We do not rule out this mechanism but note that it is not supported by the current data. Discussion of this has been added to lines 364-371.

3) There is not a consensus that PE is required for optimal activity of the nAChR (lines 89-90). The most compelling data is that anionic lipids and cholesterol are important.

While the bulk of evidence examining lipid modulation in the nAChR shows significant effects on anionic lipids and cholesterol, most of these studies are not examining ion conduction/flux but rather other measures of channel function and structure. We chose to reference this study

because it is one of the few studies, that we are aware of, that reports lipid dependence of ion flux, since we are also measuring this endpoint.

4) Line 105, if it is not statistically significant, then it is within the error of the measurement – remove “2x higher”.

Agreed. As noted above, presentation and interpretation of the EC50 data has been changed (lines 110-112).

5) The authors conclude that PG stabilizes the open state because it facilitates activation of ELIC. The electrophysiological data showing that WT ELIC desensitizes rapidly refutes this. As noted above, PG facilitates ELIC transitioning to open and then desensitized states.

This final point has been addressed above.

REVIEWERS' COMMENTS

Reviewer #1 (Remarks to the Author):

I am satisfied to the authors' responses and edits to my and other reviewers' comments.

Reviewer #2 (Remarks to the Author):

I believe the comments of all reviewer's have been addressed with the revised manuscript. I have no new or additional comments.

Reviewer #3 (Remarks to the Author):

The revision is worthy of publication in my view.